# Price equation captures the role of drug interactions and collateral effects in the evolution of multidrug resistance

**Erida Gjini[1]\*, Kevin B Wood[2]\***

[1]Center for Computational and Stochastic Mathematics, Instituto Superior Tecnico, University of Lisbon, Portugal, Lisbon, Portugal; [2]Departments of Biophysics and Physics, University of Michigan, Ann Arbor, United States

**Abstract** Bacterial adaptation to antibiotic combinations depends on the joint inhibitory effects of the two drugs (drug interaction [DI]) and how resistance to one drug impacts resistance to the other (collateral effects [CE]). Here we model these evolutionary dynamics on two-dimensional phenotype spaces that leverage scaling relations between the drug-response surfaces of drug-sensitive (ancestral) and drug-resistant (mutant) populations. We show that evolved resistance to the component drugs – and in turn, the adaptation of growth rate – is governed by a Price equation whose covariance terms encode geometric features of both the two-drug-response surface (DI) in ancestral cells and the correlations between resistance levels to those drugs (CE). Within this framework, mean evolutionary trajectories reduce to a type of weighted gradient dynamics, with the drug interaction dictating the shape of the underlying landscape and the collateral effects constraining the motion on those landscapes. We also demonstrate how constraints on available mutational pathways can be incorporated into the framework, adding a third key driver of evolution. Our results clarify the complex relationship between drug interactions and collateral effects in multidrug environments and illustrate how specific dosage combinations can shift the weighting of these two effects, leading to different and temporally explicit selective outcomes.

**\*For correspondence:**
erida.gjini@tecnico.ulisboa.pt
(EG);
kbwood@umich.edu (KBW)

**Competing interests:** The authors declare that no competing interests exist.

## Introduction

Understanding and predicting evolutionary dynamics is an ongoing challenge across all fields of biology. Microbial populations offer relatively simple model systems for investigating adaptation on multiple length scales, from the molecular to the population level, on timescales ranging from a few generations to decades. Yet even these simplest of systems exhibit rich and often counterintuitive evolutionary dynamics. Despite enormous progress, both theoretical and experimental, predicting evolution remains exceedingly difficult, in part because it is challenging to identify the phenotypes, selective gradients, and environmental factors shaping adaptation. In turn, controlling those dynamics – for example, by judicious manipulation of environmental conditions – is often impossible. These challenges represent open theoretical questions but also underlie practical public health threats – exemplified by the rapid rise of antibiotic resistance (*Davies and Davies, 2010*; *Levy and Marshall, 2004*) – where evolutionary dynamics are fundamental to the challenge, and perhaps, to the solution.

In recent years, evolution-based strategies for impeding drug resistance have gained significant attention. These approaches have identified a number of different factors that could modulate resistance evolution, including spatial heterogeneity (*Zhang et al., 2011*; *Baym et al., 2016a*; *Greulich et al., 2012*; *Hermsen et al., 2012*; *Moreno-Gamez et al., 2015*; *Gokhale et al., 2018*; *De Jong and Wood, 2018*; *Santos-Lopez et al., 2019*); competitive (*Read et al., 2011*;

*Hansen et al., 2017*; *Hansen et al., 2020*), cooperative (*Meredith et al., 2015b*; *Artemova et al., 2015*; *Sorg et al., 2016*; *Tan et al., 2012*; *Karslake et al., 2016*; *Yurtsev et al., 2016*; *Hallinen et al., 2020*), or metabolic (*Adamowicz et al., 2018*; *Adamowicz et al., 2020*) interactions between bacterial cells; synergy with the immune system, especially in the context of adaptive treatment (*Gjini and Brito, 2016*); epistasis between resistance mutations (*Trindade et al., 2009*; *Borrell et al., 2013*; *Lukačišinová et al., 2020*); plasmid dynamics (*Lopatkin et al., 2016*; *Lopatkin et al., 2017*; *Cooper et al., 2017*); precise tuning of drug doses (*Lipsitch and Levin, 1997*; *Yoshida et al., 2017*; *Meredith et al., 2015a*; *Nichol et al., 2015*; *Fuentes-Hernandez et al., 2015*; *Coates et al., 2018*; *Iram et al., 2021*); cycling or mixing drugs at the hospital level (*Bergstrom et al., 2004*; *Beardmore et al., 2017*); and statistical correlations between resistance profiles for different drugs (*Imamovic and Sommer, 2013*; *Kim et al., 2014*; *Pál et al., 2015*; *Barbosa et al., 2017*; *Rodriguez de Evgrafov et al., 2015*; *Nichol et al., 2019*; *Podnecky et al., 2018*; *Imamovic et al., 2018*; *Barbosa et al., 2019*; *Rosenkilde et al., 2019*; *Apjok et al., 2019*; *Maltas and Wood, 2019*; *Maltas et al., 2020*; *Hernando-Amado et al., 2020*; *Roemhild et al., 2020*; *Ardell and Kryazhimskiy, 2020*).

Drug combinations are a particularly promising approach for slowing resistance (*Baym et al., 2016b*), but the evolutionary impacts of combination therapy remain difficult to predict, especially in a clinical setting (*Podolsky, 2015*; *Woods and Read, 2015*). Antibiotics are said to interact when the combined effect of the drugs is greater than (synergy) or less than (antagonism) expected based on the effects of the drugs alone (*Greco et al., 1995*). These interactions may be leveraged to improve treatments – for example, by offering enhanced antimicrobial effects at reduced concentrations. But these interactions can also accelerate, reduce, or even reverse the evolution of resistance (*Chait et al., 2007*; *Michel et al., 2008*; *Hegreness et al., 2008*; *Pena-Miller et al., 2013*; *Dean et al., 2020*), leading to tradeoffs between short-term inhibitory effects and long-term evolutionary potential (*Torella et al., 2010*). In addition, resistance to one drug may be associated with modulated resistance to other drugs. This cross-resistance (or collateral sensitivity) between drugs in a combination has also been shown to significantly modulate resistance evolution (*Barbosa et al., 2018*; *Rodriguez de Evgrafov et al., 2015*; *Munck et al., 2014*).

Collateral effects (*Pál et al., 2015*; *Roemhild et al., 2020*) and drug interactions (*Bollenbach et al., 2009*; *Chevereau et al., 2015*; *Lukačišin and Bollenbach, 2019*; *Chevereau et al., 2015*), even in isolation, reflect interactions – between genetic loci, between competing evolutionary trajectories, between chemical stressors – that are often poorly understood at a mechanistic or molecular level. Yet adaptation to a drug combination may often reflect both phenomena, with the pleiotropic effects that couple resistance to individual drugs constraining, or constrained by, the interactions that occur when those drugs are used simultaneously. In addition, the underlying genotype space is high-dimensional and potentially rugged, rendering the genotypic trajectories prohibitively complex (*de Visser and Krug, 2014*).

In this work, we attempt to navigate these obstacles by modeling evolutionary dynamics on lower-dimensional phenotype spaces that leverage scaling relations between the drug-response surfaces of ancestral and mutant populations. Our approach is inspired by the fact that multiobjective evolutionary optimization may occur on surprisingly low-dimensional phenotypic spaces (*Shoval et al., 2012*; *Hart et al., 2015*). To develop a similarly coarse-grained picture of multidrug resistance, we associate selectable resistance traits with changes in effective drug concentrations, formalizing the geometric rescaling assumptions originally pioneered in *Chait et al., 2007*; *Hegreness et al., 2008*; *Michel et al., 2008* and connecting evolutionary dynamics with a simple measurable property of ancestral populations. We show that evolved resistance to the component drugs – and in turn, the adaptation of growth rate – is governed by a Price equation whose covariance terms encode geometric features of both (1) the two-drug-response surface in ancestral populations (the drug interaction) and (2) the correlations between resistance levels to those drugs (collateral effects). In addition, we show how evolutionary trajectories within this framework reduce to a type of weighted gradient dynamics on the two-drug landscape, with the drug interaction dictating the shape of the underlying landscape and the collateral effects constraining the motion on those landscapes, leading to deviations from a simple gradient descent. We also illustrate two straightforward extensions of the basic framework, allowing us to investigate the effects of both constrained mutational pathways and sequential multidrug treatments. Our results clarify the complex relationship between drug interactions and collateral effects in multidrug environments and illustrate

how specific dosage combinations can shift the weighting of these two effects, leading to different selective outcomes even when the available genetic routes to resistance are unchanged.

## Results

Our goal is to understand evolutionary dynamics of a cellular population in the presence of two drugs, drug 1 and drug 2. These dynamics reflect a potentially complex interplay between drug interactions and collateral evolutionary tradeoffs, and our aim is to formalize these effects in a simple model. To do so, we assume that the per capita growth rate of the ancestral population is given by a function $G(x, y)$, where $x$ and $y$ are the concentrations of drugs 1 and 2, respectively. We limit our analysis to two-drug combinations, but it could be extended to higher-order drug combinations, though this would require empirical or theoretical estimates for higher-dimensional drug-response surfaces (see, e.g., *Wood et al., 2012*; *Zimmer et al., 2016*; *Zimmer et al., 2017*; *Russ and Kishony, 2018*; *Tekin et al., 2016*; *Tekin et al., 2017*; *Tekin et al., 2018*). At this stage, we do not specify the functional form of $G(x, y)$, though we assume that this function can be derived from pharmacodynamic or mechanistic considerations (*Engelstädter, 2014*; *Bollenbach et al., 2009*; *Wood and Cluzel, 2012*) or otherwise estimated from experimental data (*Greco et al., 1995*; *Wood et al., 2014*). In classical pharmacology models (*Loewe, 1953*; *Greco et al., 1995*), the shape of these surfaces – specifically, the convexity of the corresponding contours of constant growth ('isoboles') – determines the type of drug interaction, with linear isoboles corresponding to additive drug pairs. In this framework, deviations from linearity indicate synergy (concave up) or antagonism (concave down). While there are multiple conventions for assigning geometric features of the response surface to an interactions type – and there has been considerable debate about the appropriate null model for additive interactions (*Greco et al., 1995*) – the response surfaces contain complete information about the phenotypic response. The manner in which this response data is mapped to a qualitative interaction type – and therefore used to label the interaction as synergistic, for example – is somewhat subjective, though in what follows we adopt the isobole-based labeling scheme because it is more directly related to the geometry of the response surface than competing models (e.g., Bliss independence; *Greco et al., 1995*).

### Resistance as a continuous trait and rescaling in a simple model

The primary assumption of the model is that the phenotypic response (e.g., growth rate) of drug-resistant mutants, which may be present in the initial population or arise through mutation, corresponds to a simple rescaling of the growth rate function $G(x, y)$ for the ancestral population. As we will see, this scheme provides an explicit link between a cell's level of antibiotic resistance and its fitness in a given multidrug environment. Specifically, we assume that the growth rate ($g_i$) of mutant $i$ is given by

$$g_i = G(\alpha_i x, \beta_i y), \tag{1}$$

where $\alpha_i$ and $\beta_i$ are rescaling parameters that reflect the physiological effects of mutations on the growth rate. In some cases – for example, resistance due to efflux pumps (*Wood and Cluzel, 2012*) or drug degrading enzymes (*Yurtsev et al., 2013*) – this effective concentration change corresponds to a physical change in intracellular drug concentration. More generally, though, this hypothesis assumes that resistant cells growing in external drug concentration $x$ behave similarly to wild-type (drug-sensitive) cells experiencing a reduced effective concentration $\alpha x$. Similar rescaling arguments were originally proposed in *Chait et al., 2007*, where they were used to predict correlations between the rate of resistance evolution and the type of drug interaction. These arguments have also been used to describe fitness tradeoffs during adaptation (*Das et al., 2020*) and to account for more general changes in the dose-response curves, though in many cases the original two-parameter rescaling was sufficient to describe the growth surface in mutants (*Wood et al., 2014*).

When only a single drug is used, this rescaling leads to a simple relationship between the characteristic inhibitory concentrations – for example, the half-maximal inhibitory concentration (IC$_{50}$) or the minimum inhibitory concentration (MIC) – of the ancestral (sensitive) and mutant (resistant) populations. In what follows, we refer to these reference concentrations as $K_i^j$, where $i$ labels the cell type and, when there is more than one drug, $j$ labels the drug. Conceptually, this means that dose-

response curves for both populations have the same basic shape, with resistance (or sensitivity) in the mutant corresponding only to a shape-preserving rescaling of the drug concentration ($D \to D/K_i$; *Figure 1A*). In the presence of two drugs, the dose-response curves become dose-response surfaces, and rescaling now corresponds to a shape-preserving rescaling of the contours of constant growth. There are now two scaling parameters, one for each drug, and in general they are not equal. For example, in *Figure 1B*, the mutant shows increased sensitivity to drug 1 ($\alpha \equiv K_{\mathrm{WT}}^1/K_{\mathrm{Mut}}^1 > 1$) and increased resistance to drug 2 ($\beta \equiv K_{\mathrm{WT}}^2/K_{\mathrm{Mut}}^2 < 1$), where superscripts label the drug (1 or 2) and subscripts label the cell type (wild type, WT; mutant, Mut).

The power of this rescaling approach is that it directly links growth of the mutant populations to measurable properties of the ancestral population (the two-drug-response surface) via traits of the mutants (the rescaling parameters). Each mutant is characterized by a pair of scaling parameters, ($\alpha_i, \beta_i$), which one might think of as a type of coarse-grained genotype (*Figure 1C*). When paired with the ancestral growth surface, these traits fully determine the per capita growth rate of the mutant at any external dosage combination ($x, y$) via *Equation 1*. While the scaling parameters are intrinsic properties of each mutant, they contribute to the phenotype (growth) in a context-dependent manner, leading to selection dynamics that depend in predictable ways on the external environment (*Figure 1*).

We assume a finite set of $M$ subpopulations (mutants), ($i = 1, \ldots M$), with each subpopulation corresponding to a single pair of scaling parameters. For simplicity, initially we assume each of these mutants is present in the original population at low frequency and neglect mutations that could give rise to new phenotypes, though later we show that it is straightforward to incorporate them into the same framework. We do not specify the mechanisms by which this standing variation is initially generated or maintained; instead, we simply assume that such standing variation exists, and our goal is to predict how selection will act on this variation for different choices of external (true) drug concentrations. As we will see, statistical properties of this variation combine with the local geometry of the response surface to determine the selection dynamics of these traits.

## Population dynamics of scaling parameters

The mean resistance trait to drug 1, which in our case is the scaling parameter $\bar{\alpha}(t) \equiv \sum_{i=1}^{M} \alpha_i f_i(t)$, evolves according to

$$\frac{d\bar{\alpha}}{dt} = \sum_{i=1}^{M} \alpha_i \frac{df_i(t)}{dt}, \tag{2}$$

where $f_i(t)$ is the population frequency of mutant $i$ at time $t$ in the population. Assuming that each subpopulation grows exponentially at the per capita growth rate ($dn_i/dt = g_i n_i$, with $n_i$ the abundance of mutant $i$ and $g_i$ the per capita growth rate given by *Equation 1*), the frequency $f_i(t)$ changes according to

$$\frac{df_i}{dt} = \frac{d}{dt}\left(\frac{n_i}{\sum_i n_i}\right) = f_i(g_i - \bar{g}), \tag{3}$$

where $\bar{g} = \sum_{i=1}^{M} f_i g_i$ is the (time-dependent) mean value of $g_i$ across all $M$ subpopulations (mutants). Combining *Equations 2 and 3*, we have

$$\frac{d\bar{\alpha}}{dt} = \mathrm{Cov}_{\boldsymbol{x}}(\alpha, g), \tag{4}$$

where $\mathrm{Cov}(\alpha, g)_{\boldsymbol{x}} \equiv \sum_{i=1}^{M} \alpha_i f_i(g_i - \bar{g})$ is the covariance between the scaling parameters $\alpha_i$ and the corresponding mutant growth rates $g_i$. The subscript $\boldsymbol{x}$ is a reminder that the growth rates $g_i$ and $\bar{g}$ that appear in the covariance sum depend on the external (true) drug concentration $\boldsymbol{x} \equiv (x, y)$. An identical derivation leads to an analogous equation for the scaling parameter with respect to drug 2, mean susceptibility to drug 2, $\bar{\beta}$, relative to the original population, and the full dynamics are therefore described by

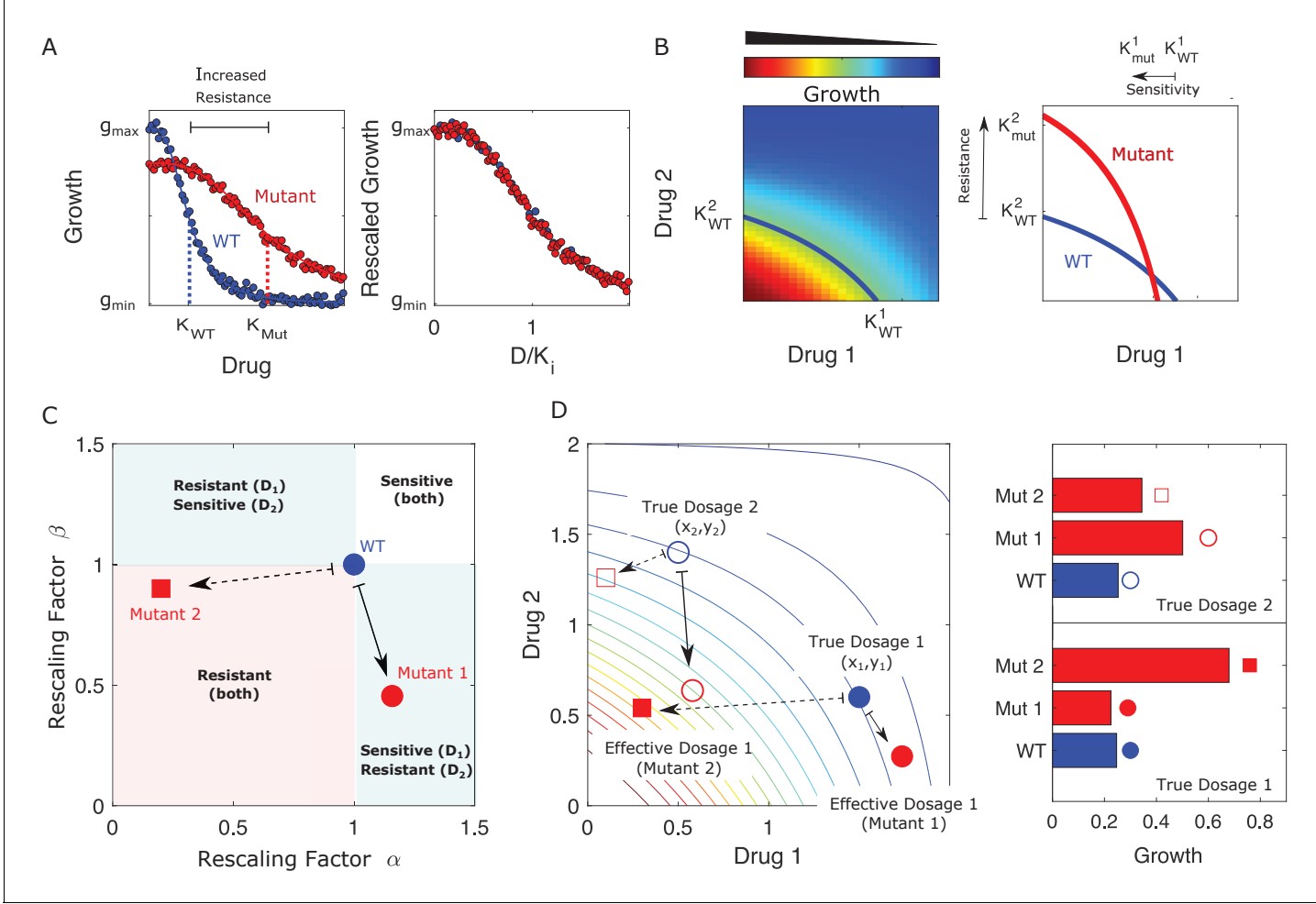

**Figure 1.** Drug resistance as a rescaling of effective drug concentration. The fundamental assumption of our model is that drug-resistant mutants exhibit phenotypes identical to those of the ancestral ('wild type') cells but at rescaled effective drug concentration. (A) Left panel: schematic dose-response curves for an ancestral strain (blue) and a resistant mutant (red). Half-maximal inhibitory concentrations ($K_{\text{WT}}, K_{\text{Mut}}$), which provide a measure of resistance, correspond to the drug concentrations where inhibition is half maximal. Fitness cost of resistance is represented as a decrease in drug-free growth. Right panel: dose-response curves for both cell types collapse onto a single functional form, similar to those in *Chait et al., 2007*; *Michel et al., 2008*; *Wood and Cluzel, 2012*; *Wood et al., 2014*. (B) Left panel: in the presence of two drugs, growth is represented by a surface; the thick blue curve represents the isogrowth contour at half-maximal inhibition; it intersects the axes at the half-maximal inhibitory concentrations for each individual drug. Right panel: isogrowth contours for ancestral (WT) and mutant cells. In this example, the mutant exhibits increases resistance to drug 2 and an increased sensitivity to drug 1, each of which corresponds to a rescaling of drug concentration for that drug. These rescalings are quantified with scaling constants $\alpha \equiv K^1_{\text{WT}}/K^1_{\text{Mut}}$ and $\beta \equiv K^2_{\text{WT}}/K^2_{\text{Mut}}$, where the superscripts indicate the drug (1 or 2). (C) Scaling factors for two different mutants (red square and red circle) are shown. The ancestral cells correspond to scaling constants $\alpha = \beta = 1$. Mutant 1 exhibits increased sensitivity to drug 1 ($\alpha > 1$) and increased resistance to drug 2 ($\beta < 1$). Mutant 2 exhibits increased resistance to both drugs ($\alpha < \beta < 1$), with higher resistance to drug 1. (D) Scaling parameters describe the relative change in effective drug concentration experienced by each mutant. While scaling parameters for a given mutant are fixed, the effects of those mutations on growth depend on the external environment (i.e., the drug dosage applied). This schematic shows the effective drug concentrations experienced by WT cells (blue circles) and the two different mutants (red circles and red squares) from panel (C) under two different external conditions (open and closed shapes). True dosage 1 (2) corresponds to higher external concentrations of drug 1 (2). The concentrations are superimposed on a contour plot of the two drug surface (similar to panel B). Right panel: resulting growth of mutants and WT strains at dosage 1 (bottom) and dosage 2 (top). Because the dosages are chosen along a contour of constant growth, the WT exhibits the same growth at both dosages. However, the growth of the mutants depends on the dose, with mutant 1 growing faster (slower) than mutant 2 under dosage 2 (dosage 1). A key simplifying feature of these evolutionary dynamics is that the selective regime (drug concentration) and phenotype (effective drug concentration) have same units.

The online version of this article includes the following video for figure 1:

**Figure 1—video 1.** Detailed selection dynamics associated to *Figure 2*.

https://elifesciences.org/articles/64851#fig1video1

$$\frac{d\bar{\alpha}}{dt} = \text{Cov}_{\boldsymbol{x}}(\alpha, g),$$
$$\frac{d\bar{\beta}}{dt} = \text{Cov}_{\boldsymbol{x}}(\beta, g). \tag{5}$$

To complete the model described by *Equation 5*, one must specify the external (true) concentration of each drug ($\boldsymbol{x}$); a finite set of scaling parameter pairs $\alpha_i, \beta_i$ corresponding to all 'available' mutations; and an initial condition ($\bar{\alpha}(0), \bar{\beta}(0)$) for the mean scaling parameters. When combined with the external drug concentrations, the scaling parameters directly determine the effective drug concentrations ($D_1^{\text{eff}}, D_2^{\text{eff}}$) experienced by each mutant according to

$$D_1^{\text{eff}} = \alpha_i x,$$
$$D_2^{\text{eff}} = \beta_i y. \tag{6}$$

We note that drug concentrations can be above or below the MIC contour, with higher concentrations leading to population collapse ($G<0$) and lower concentrations to population growth ($G>0$) in the ancestral (drug sensitive) cells. However selection dynamics remain the same in both cases as selection depends only on differences in growth rates between different subpopulations.

*Equation 5* is an example of the well-known Price equation from evolutionary biology (*Price, 1970*; *Price, 1972*; *Frank, 1995*; *Lehtonen et al., 2020*), which says that the change in the (population) mean value of a trait is governed by the covariance of traits and fitness. In general, fitness can be difficult to measure and, in some cases, even difficult to define. However, the rescaling assumption of our model replaces fitness with $g$, which can be directly inferred from measurable properties (the two-drug-response surface) in the ancestral population. In what follows, we will sometimes casually refer to *Equation 5* as a 'model,' but it is important to note that the Price equation is not, in and of itself, a mathematical model in the traditional sense. Instead, it is a simple mathematical statement describing the statistical relationship between variables, which are themselves defined in some underlying model. In this case, the mathematical model consists of a collection of exponentially growing populations whose per capita growth rates are linked by scaling relationships. *Equation 5*– the Price equation – does not include additional assumptions, mechanistic or otherwise, but merely captures statistical relationships between the model variables. We will see, however, that these relationships provide conceptual insight into the interplay between collateral effects and drug interactions.

*Equation 5* encodes deceptively rich dynamics that depend on both the interaction between drugs and the collateral effects of resistance. First, it is important to note that αs and βs vary together in pairs, and the evolution of these two traits is not independent. As a result, constraints on the joint co-occurrence of $\alpha_i$ and $\beta_i$ among the mutant subpopulations can significantly impact the dynamics. These constraints correspond to correlations between resistance levels to different drugs – that is, to cross-resistance (when pairs of scaling parameters simultaneously increase resistance to both drugs) or to collateral sensitivity (when one scaling parameter leads to increased resistance and the other to increased sensitivity). In addition, $g$ contains information about the dose-response surface and, therefore, about the interaction between drugs. The evolution of the scaling parameters is not determined solely by the drug interaction or by the collateral effects, but instead by both – quantified by the covariance between these rescaled trait values and the ancestral dose-response surface.

As an example, we integrated the model numerically to determine the dynamics of the mean scaling parameters and the mean growth rate for a population exposed to a fixed concentration of two drugs whose growth surface has been fully specified (*Figure 2A*). The dynamics can be thought of as motion on the two-dimensional response surface; if the initial population is dominated by the ancestral cells, the mean scaling parameters are approximately 1, and the trajectory therefore starts near the point representing the true drug concentration (in this case, $(x_0, y_0)$), where growth is typically small (*Figure 2A*). Over time, the mean traits evolve, tracing out a trajectory in the space of scaling parameters (*Figure 2B*). When the external concentration of drug is specified, these dynamics also correspond to a trajectory through the space of effective drug concentrations, which, in turn, can be linked with an average growth rate through the drug-response surface (*Figure 2B*). The model therefore describes both the dynamics of the scaling parameters, which describe how resistance to each

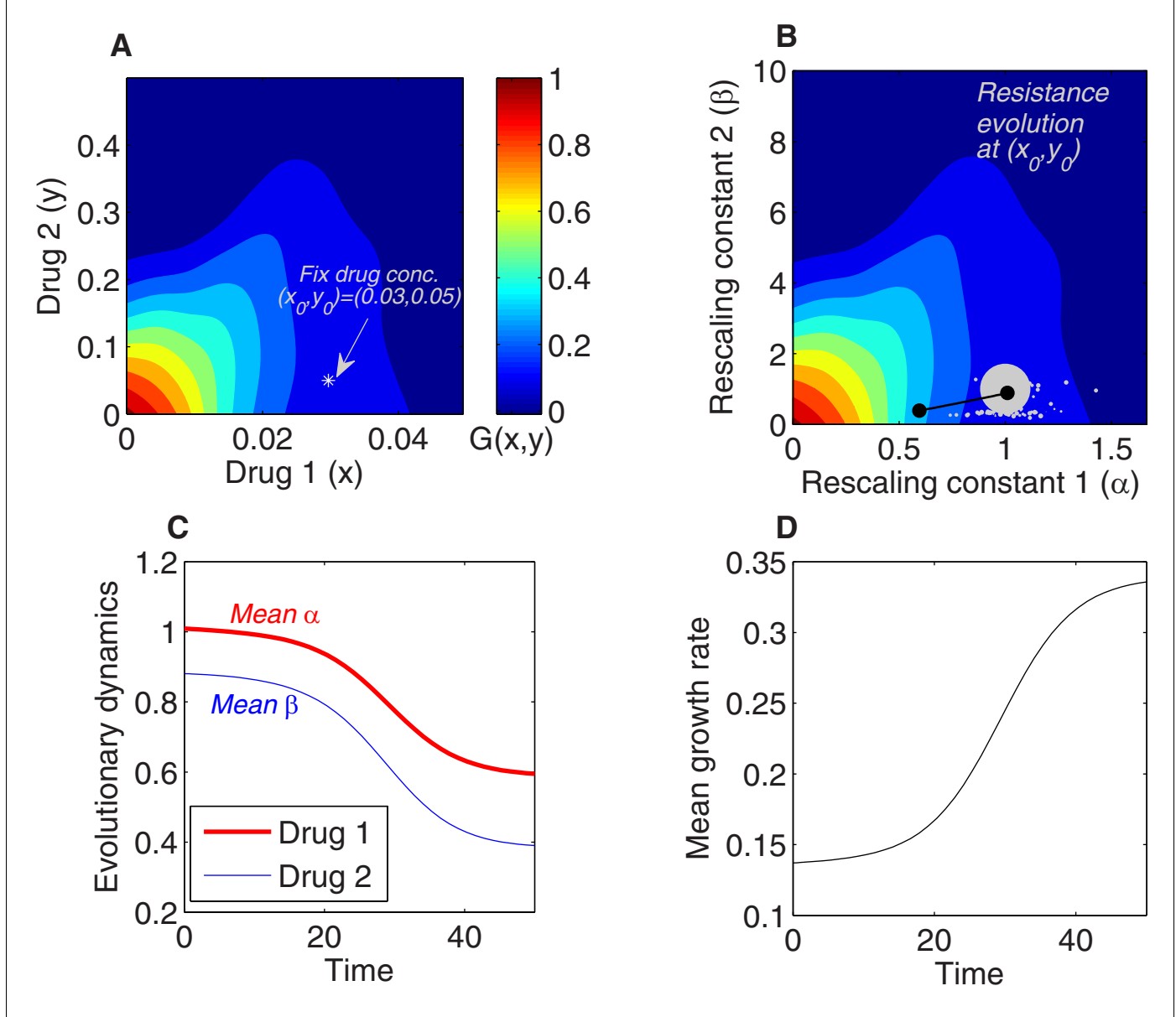

**Figure 2.** Using the rescaling framework for predicting resistance evolution to two drugs in a population of bacterial cells. (A) Growth landscape (per capita growth rate relative to untreated cells) as a function of drug concentration for two drugs (drug 1, tigecycline; drug 2, ciprofloxacin; concentrations measured in µg/mL) based on measurements in *Dean et al., 2020*. We consider evolution of resistance in a population exposed to a fixed external drug concentration $(x_0, y_0) = (0.03, 0.05)$ (white asterisk). (B) Growth landscape from (A) with axes rescaled to reflect scaling parameters $(\alpha, \beta)$. The point $(x_0, y_0)$ in drug concentration space now corresponds to the point $(1, 1)$ in scaling parameter space; intuitively, a strain characterized by scaling parameters of unity experiences an effective drug concentration equal to the true external concentration. We consider a population that is primarily ancestral cells (fraction 0.99), with the remaining fraction uniformly distributed between an empirically measured collection of mutants (each corresponding to a single pair of scaling parameters, white circles). At time 0, the population is primarily ancestral cells and is therefore centered very near $(1, 1)$ (gray circle). Over time, the mean value of each scaling parameter decreases as the population becomes increasingly resistant to each drug (black curve). (C) The two-dimensional mean scaling parameter trajectory from (B) plotted as two time series, one for $\alpha$ (red) and one for $\beta$ (blue). For the detailed selection dynamics, see *Figure 1—video 1*. (D) The mean fitness of the population during evolution is expected to increase and can be computed from the selection dynamics by numerically integrating the model at each time step.

drug changes over time (*Figure 2C*), and the dynamics of growth rate adaptation in the population as a whole (*Figure 2D*).

## Selection dynamics depend on drug interaction and collateral effects

This rescaling model indicates that selection is determined by both the drug interaction and the collateral effects, consistent with previous experimental findings. As a simple example, consider the case of a fixed drug interaction but different qualitative types of collateral effects – that is, different statistical relationships between resistance to drug 1 (via $\alpha$) and resistance to drug 2 (via $\beta$). In *Figure 3A*, we consider cases where resistance is primarily to drug 2 (black), primarily to drug 1 (cyan), strongly positively correlated (cross-resistance, pink), and strongly negatively correlated (collateral sensitivity, green). Using the same drug interaction surface as in *Figure 2*, we find that a mixture of both drugs leads to significantly different trajectories in the space of scaling parameters (*Figure 3B*) and, in turn, significantly different rates of growth adaptation. In this example, cross-resistance (pink) leads to rapid increases in resistance to both drugs (rapid decreases in scaling parameters, *Figure 3C, D*) and the fastest growth adaptation (*Figure 3E*). By contrast, if resistance is limited primarily to one drug (cyan or black), growth adaptation is slowed (*Figure 3E*) – intuitively, purely horizontal or purely vertical motion in the space of scaling parameters leads to only a modest change in growth because the underlying response surface is relatively flat in those directions, meaning that the rescaled concentration, in each case, lies near the original contour (*Figure 3*). As a result of the contour shapes (drug interaction), resistance to both drugs can develop at approximately the same rate (for example), even when collateral structure suggests resistance to one drug will dominate (e.g., cyan case in *Figure 3*). We note that the dynamics will depend on the (true) external drug concentrations $(x_0, y_0)$, even if the rescaling parameters remain identical, because a given rescaling transformation will lead to different effective drug concentrations, and therefore different growth rates, for different values of $(x_0, y_0)$.

When both collateral effects and drug interactions vary, the dynamics can be considerably more complex, and the dominant driver of adaptation can be drug interaction, collateral effects, or a combination of both. Previous studies support this picture as adaptation has been observed to be driven primarily by drug interactions (*Chait et al., 2007*; *Michel et al., 2008*; *Hegreness et al., 2008*), primarily by collateral effects (*Munck et al., 2014*; *Barbosa et al., 2018*), or by combinations of both (*Baym et al., 2016b*; *Barbosa et al., 2018*; *Dean et al., 2020*). *Figure 4* shows schematic examples of growth rate adaptation for different types of collateral effects (rows, ranging from cross-resistance [top] to collateral sensitivity [bottom]) and drug interactions (columns, ranging from synergy [left] to antagonism [right]). The growth adaptation may also depend sensitively on the external environment – that is, on the true external drug concentration (blue, cyan, and red). In the absence of both drug interactions (linear isoboles) and collateral effects (uncorrelated scaling parameters), adaptation is slower when the drugs are used in combination than when they are used alone, consistent with the fact that adaptation to multiple stressors is expected to take longer than adaptation to each stressor alone (*Figure 4*; middle row, middle column). As in *Figure 3*, modulating collateral effects with drug interaction fixed can have dramatic impacts on adaptation (*Figure 4*, columns). On the other hand, modulating the drug interaction in the absence of collateral effects will also significantly impact adaptation, with synergistic interactions leading to accelerated adaptation relative to other types of drug interaction (*Figure 4*, middle row; compare green curves across row). Similar interaction-driven adaptation has been observed in multiple bacterial species (*Chait et al., 2007*; *Michel et al., 2008*; *Hegreness et al., 2008*; *Dean et al., 2020*).

## Selection as weighted gradient dynamics on the ancestral response surface

To gain intuition about the dynamics in the presence of both collateral effects and drug interactions, we consider an approximately monomorphic population where scaling parameters are initially narrowly distributed around their mean values. In this case, we can Taylor expand the function $g_i = G(\alpha_i x_0, \beta_i y_0)$ (corresponding to the growth of mutant $i$) around the mean values $\bar{\alpha}, \bar{\beta}$, leading to

$$g_i \approx G(\bar{\alpha} x_0, \bar{\beta} y_0) + \partial_x G(x, y)_{\bar{\alpha} x_0, \bar{\beta} y_0} (\alpha_i - \bar{\alpha}) x_0 + \partial_y G(x, y)_{\bar{\alpha} x_0, \bar{\beta} y_0} (\beta_i - \bar{\beta}) y_0, \qquad (7)$$

where we have neglected terms quadratic and higher. In this regime, $g_i$ is a linear function of the scaling parameters, and the covariances can therefore be written as

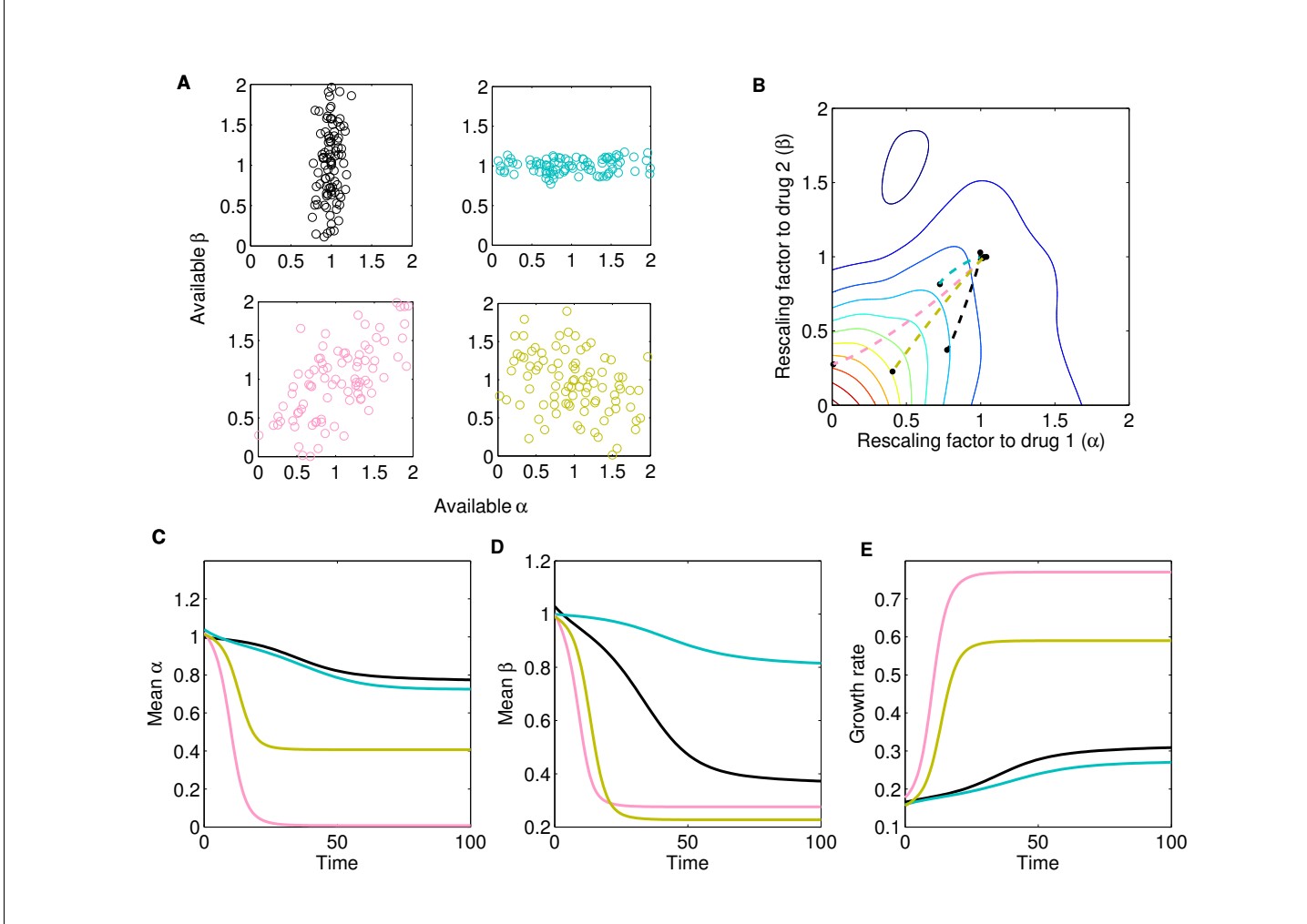

**Figure 3.** Different collateral profiles drive different evolutionary dynamics under the same treatment. We simulated random collateral profiles for susceptibilities to two drugs and use them to predict phenotypic trajectories of a bacterial population. For the two-drug landscape, we chose one of the experimentally measured surfaces from *Dean et al., 2020; Figure 3—figure supplement 1*C, corresponding to the drug combination TGC-CIP (drug 1-drug 2). (A) Four special profiles include predominant variation in β (black), predominant variation in α (cyan), positive correlation between αs and βs (pink), and negative correlation between the susceptibilities to the two drugs (green). These were generated from a bivariate normal distribution with mean $(1, 1)$ and covariance matrices $\Sigma_1 = \begin{pmatrix} 0.01 & 0.02 \\ 0.02 & 0.4 \end{pmatrix}$, $\Sigma_2 = \begin{pmatrix} 0.5 & 0.02 \\ 0.02 & 0.01 \end{pmatrix}$, $\Sigma_3 = \begin{pmatrix} 0.3 & 0.21 \\ 0.21 & 0.3 \end{pmatrix}$, $\Sigma_4 = \begin{pmatrix} 0.3 & -0.15 \\ -0.15 & 0.3 \end{pmatrix}$. (B) The trajectories in mean $\alpha - \beta$ space following treatment $(x_0, y_0)$, with $x_0 = 0.5x_{max} = 0.025$ and $y_0 = 0.5y_{max} = 0.25$, corresponding to different collateral profiles. (C) The dynamics of mean susceptibility to drug 1 $\bar{\alpha}(t)$ for the four cases. (D) The dynamics of mean susceptibility to drug 2 $\bar{\beta}(t)$ for the four cases. (E) The dynamics of mean growth rate for the four cases. For underlying heterogeneity, we drew 100 random $\alpha_i$ and $\beta_i$ as shown in (A) and initialized dynamics at ancestor frequency 0.99 and the remaining 1% evenly distributed among available mutants. It is clear that each collateral structure in terms of the available $(\alpha_i, \beta_i)$ leads to different final evolutionary dynamics under the same two-drug treatment. In this particular case, the fastest increase in resistance to two drugs and increase in growth rate occurs for the collateral resistance (positive correlation) case. The time course of the detailed selective dynamics in these four cases is depicted in *Figure 3—videos 1–4*.

The online version of this article includes the following video and figure supplement(s) for figure 3:

**Figure supplement 1.** Response surfaces and scaling parameters for three drug pairs.

**Figure 3—video 1.** Evolutionary dynamics for collateral effects in *Figure 3A*.
https://elifesciences.org/articles/64851#fig3video1

**Figure 3—video 2.** Evolutionary dynamics for collateral effects in *Figure 3B*.
https://elifesciences.org/articles/64851#fig3video2

**Figure 3—video 3.** Evolutionary dynamics for collateral effects in *Figure 3C*.
https://elifesciences.org/articles/64851#fig3video3

**Figure 3—video 4.** Evolutionary dynamics for collateral effects in *Figure 3B*.

*Figure 3 continued on next page*

$$[ll]\mathrm{Cov}_{\boldsymbol{x}_0}(\alpha,g) = \sigma_{\alpha\alpha}\,x_0\partial_x G + \sigma_{\alpha\beta}\,y_0\partial_y G,$$
$$\mathrm{Cov}_{\boldsymbol{y}_0}(\beta,g) = \sigma_{\alpha\beta}\,x_0\partial_x G + \sigma_{\beta\beta}\,y_0\partial_y G, \tag{8}$$

where $\sigma_{uv} = \overline{uv} - \bar{u}\bar{v}$ and we have used the fact that $\bar{g} \equiv \sum_i f_i G(\alpha_i x_0, \beta_i x_0) = G(\bar{\alpha}x_0, \bar{\beta}y_0)$ to first order. This is a type of weak-selection approximation: the trait that is evolving may have a very strong effect on fitness, but if there is only minor variation in such trait in the population, there will only be minor differences in fitness. *Equation 5* for the rate of change in mean traits therefore reduces to

$$\frac{d\alpha}{dt} = \boldsymbol{\Sigma}\nabla G \tag{9}$$

where $\alpha$ is a vector of mean scaling factors with components $\bar{\alpha}$ and $\bar{\beta}$, $\boldsymbol{\Sigma}$ is a covariance-like matrix given by

$$\boldsymbol{\Sigma} = \begin{pmatrix} \sigma_{\alpha\alpha} & \sigma_{\alpha\beta} \\ \sigma_{\alpha\beta} & \sigma_{\beta\beta} \end{pmatrix} \tag{10}$$

and $\nabla G$ is a weighted gradient of the function $G(x,y)$ evaluated at the mean scaling parameters,

$$\nabla G = \begin{pmatrix} x_0\partial_x G(x,y) \\ y_0\partial_y G(x,y) \end{pmatrix}_{x=x_0\bar{\alpha}, y=y_0\bar{\beta}} = \begin{pmatrix} \partial_\alpha G(x_0\alpha, y_0\beta) \\ \partial_\beta G(x_0\alpha, y_0\beta) \end{pmatrix}_{\alpha=\bar{\alpha}, \beta=\bar{\beta}}. \tag{11}$$

*Equation 9* provides an accurate description of the full dynamics across a wide range of conditions (*Figure 4—figure supplement 1*) and has a surprisingly simple interpretation. Adaptation dynamics are driven by a type of weighted gradient dynamics on the response surface, with the weighting determined by the correlation coefficients describing resistance levels to the two drugs. In the absence of collateral effects – for example, when $\boldsymbol{\Sigma}$ is proportional to the identity matrix – the scaling parameters trace out trajectories of steepest ascent on the two-drug-response surface. That is, in the absence of constraints on the available scaling parameters, adaptation follows the gradient of the response surface to rapidly achieve increased fitness, and because the response surface defines the type of drug interaction, that interaction governs the rate of adaptation. On the other hand, collateral effects introduce off-diagonal elements of $\boldsymbol{\Sigma}$, biasing trajectories away from pure gradient dynamics to account for the constraints of collateral evolution.

## Model predicts experimentally observed adaptation of growth and resistance

Our model makes testable predictions for adaptation dynamics of both the population growth rate and the population-averaged resistance levels to each drug (i.e., the mean scaling parameters) for a given drug-response surface, a given set of available mutants, and a specific combination of (external) drug dosages. To compare predictions of the model with experiment, we solved *Equation 5* for the 11 different dosage combinations of tigecycline (TGC) and ciprofloxacin (CIP) used to select drug-resistance mutants in *Dean et al., 2020*. We assumed that the initial population was dominated by ancestral cells ($\alpha = \beta = 1$) but also included a subpopulation of resistant mutants whose scaling parameters were uniformly sampled from those measured across multiple days of the evolution (see Materials and methods). The model predicts different trajectories for different external doses (selective regimes: red to blue, *Figure 5*, top panel), leading to dramatically different changes in resistance ($IC_{50}$) and growth rate adaptation (*Figure 5*, bottom panels), similar to those observed in experiment. Specifically, the model predicts (and experiments confirm) a dramatic decrease in CIP resistance as TGC concentration is increased along a contour of constant growth. As TGC eclipses acritical concentration of approximately 0.025 g/mL, selection for both CIP resistance and increased growth is eliminated. We note that comparisons between the model and experiment involve no adjustable parameters, and while the model captures the qualitative trends, it slightly but systematically underestimates the growth across TGC concentrations – perhaps suggesting additional

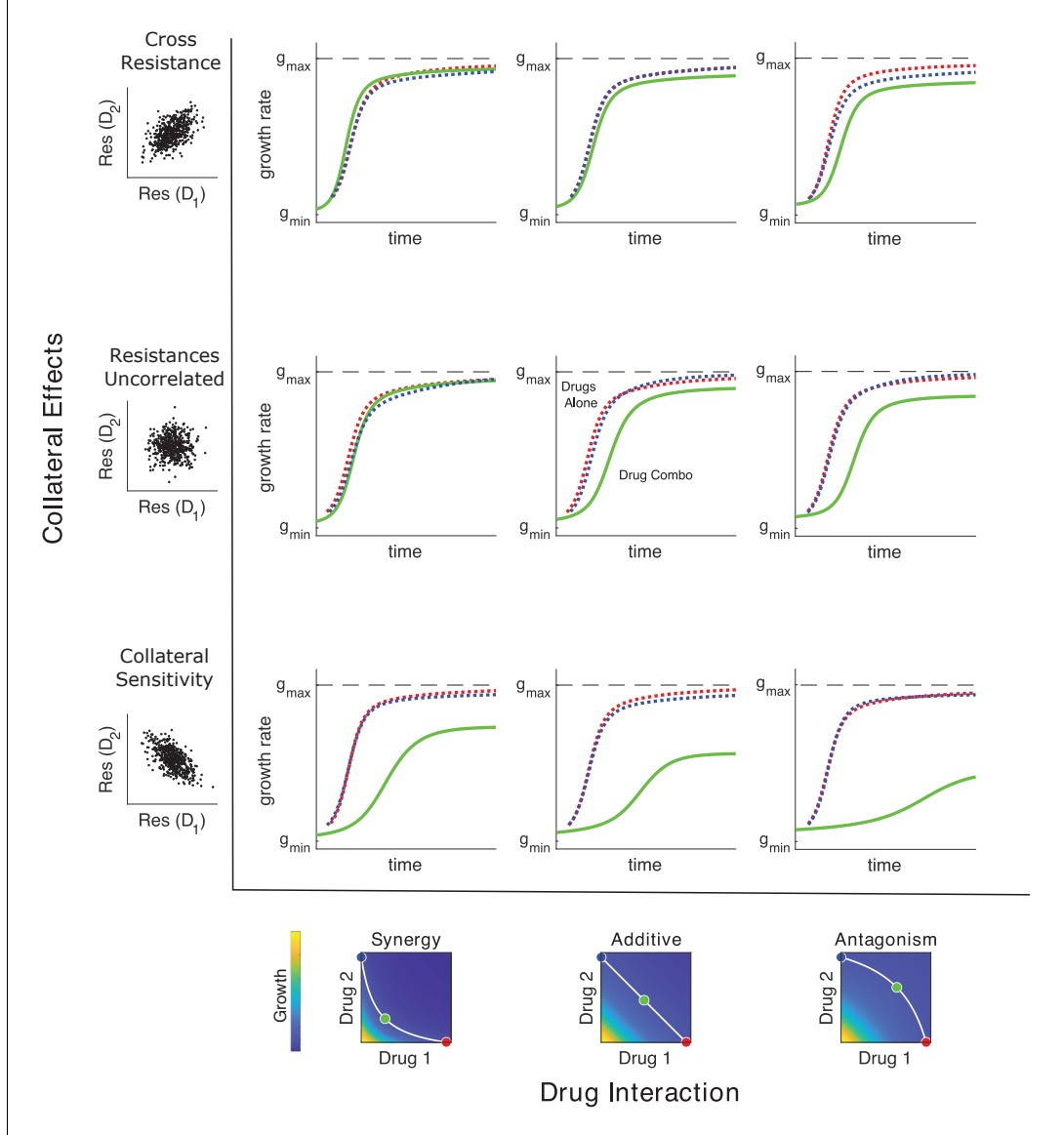

**Figure 4.** Adaptation depends on drug interactions and collateral effects of resistance. Drug interactions (columns) and collateral effects (rows) modulate the rate of growth adaptation (main nine panels). Evolution takes place at one of three dosage combinations (blue, drug 2 only; cyan, drug combination; red, drug 1 only) along a contour of constant growth in the ancestral growth response surface (bottom panels). Drug interactions are characterized as synergistic (left), additive (center), or antagonistic (right) based on the curvature of the isogrowth contours. Collateral effects describe the relationship between the resistance to drug 1 and the resistance to drug 2 in an ensemble of potential mutants. These resistance levels can be positively correlated (top row, leading to collateral resistance), uncorrelated (center row), or negatively correlated (third row, leading to collateral sensitivity). Growth adaptation (main nine panels) is characterized by growth rate over time, with dashed lines representing evolution in single drugs and solid lines indicating evolution in the drug combination. In this example, response surfaces are generated with a classical pharmacodynamic model (symmetric in the two drugs) that extends Loewe additivity by including a single-drug interaction index that can be tuned to create surfaces with different combination indices (*Greco et al., 1995*). The initial population consists of primarily ancestral cells ($\alpha = \beta = 1$) along with a subpopulation $10^{-2}$ mutants uniformly distributed among the different phenotypes. Scaling parameters are sampled from a bivariate normal distribution with equal variances ($\sigma_\alpha = \sigma_\beta$) and correlation ranging from 0.6 (top row) to $-0.6$ (bottom row). See also *Figure 4—figure supplement 1* for gradient approximation to selection dynamics.

The online version of this article includes the following figure supplement(s) for figure 4:

**Figure supplement 1.** Illustration of the local growth gradient approximation to the Price equation.

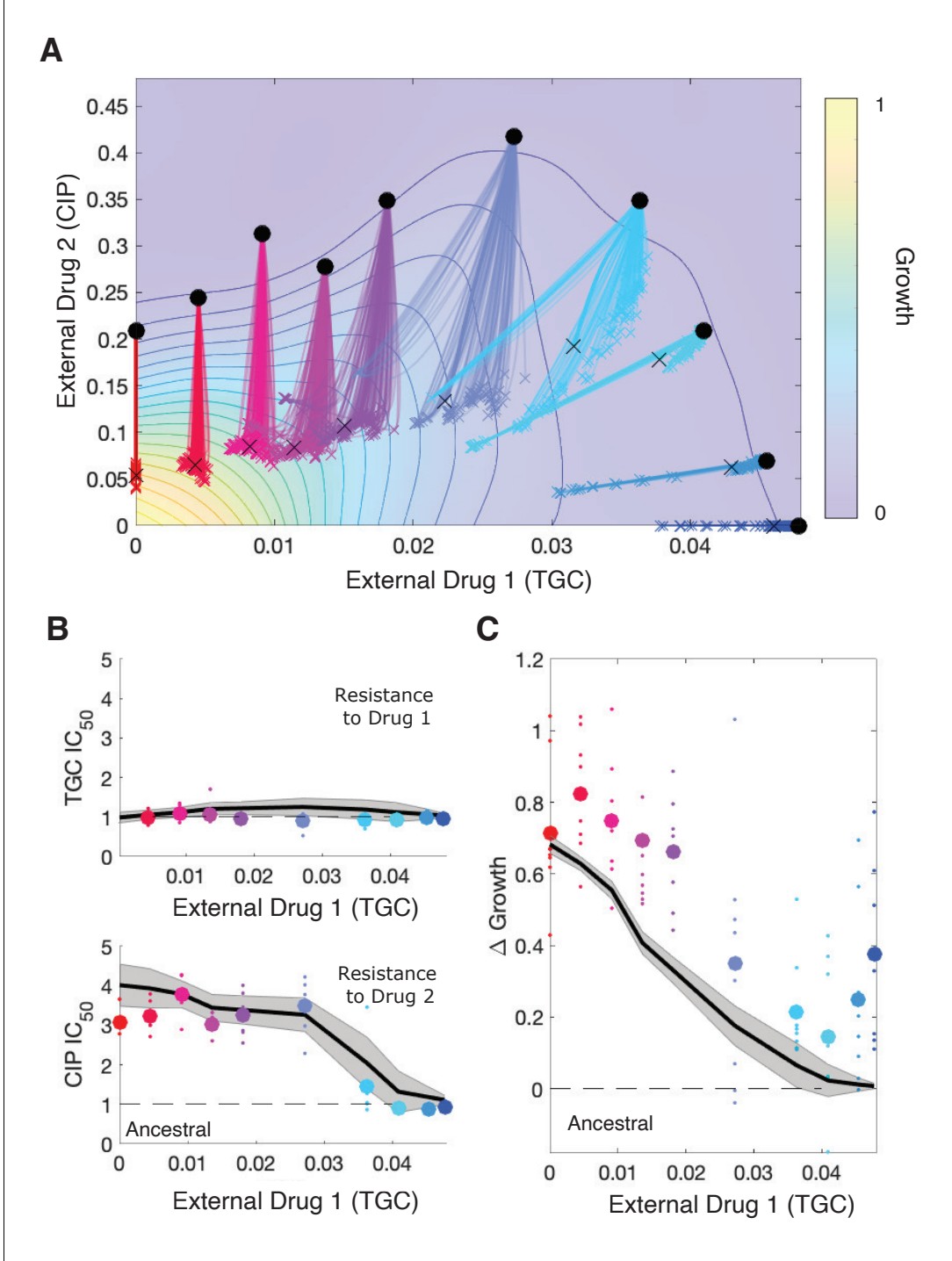

**Figure 5.** Drug rescaling describes adaptation dynamics of growth and drug resistance in *Enterococcus faecalis* under different selective regimes. (A) Experimentally measured growth response surface for tigecycline (TGC) and ciprofloxacin (CIP). Circles represent 11 different adaptation conditions, each corresponding to a specific dosage combination $(x_0, y_0)$. Solid lines show 100 simulated adaptation trajectories (i.e., changes in effective drug concentration over time: mean D1$^{eff}$ D2$^{eff}$, for a total time $T$) predicted from the full rescaling model. Black xs indicate the mean (across all trajectories) value of the effective drug concentration at the end of the simulation. In each case, the set of available mutants – and hence, the set of possible scaling parameters $\alpha_i$ and $\beta_i$ – is probabilistically determined by uniformly sampling approximately 10 scaling parameter pairs from the ensemble experimentally measured across parallel evolution experiments involving this drug pair (see Materials and methods). (B) Half-maximal inhibitory concentration (IC$_{50}$, normalized by that of the ancestral strain) for populations adapted for a fixed time period $T \approx 72$ hr in each of the 11 conditions in the top panel. The solid curve is the mean trajectory from the rescaling model (normalized IC$_{50}$ corresponds to the reciprocal of the corresponding

*Figure 5 continued on next page*

*Figure 5 continued*

scaling constant) for 100 simulated samples of the scaling parameter pairs; shaded region is ±1 standard deviation over all trajectories. Small circles are experimental observations from individual populations; larger markers are average experimental results over all populations adapted to a given condition. Note that drug 1 (TGC) concentration (x-axis) here is just a proxy for the different conditions as you move left to right along the contour in panel (A). (C) Change in population growth rate ($g_m - g_a$, where $g_m$ is per capita growth of ancestral strain and $g_a$ for the mutant; all growth rates are normalized to ancestral growth in the absence of drugs) for populations adapted in each condition. Solid curve is the mean trajectory across samplings, with shaded region ±1 standard deviation. Small circles are results from individual populations; larger markers are averages over all populations adapted to a given condition. Experimental data from *Dean et al., 2020*. See also *Figure 5—figure supplements 1–5*.

The online version of this article includes the following figure supplement(s) for figure 5:

**Figure supplement 1.** Model captures qualitative features of experimental evolution in combinations of ampicillin (AMP) and streptomycin (STR).

**Figure supplement 2.** Model captures qualitative features of experimental evolution in combinations of ceftriaxone (CRO) and ciprofloxacin (CIP).

**Figure supplement 3.** Resampling ensemble of scaling parameters does not dramatically impact dynamics.

**Figure supplement 4.** Different estimates of total time lead to similar qualitative dynamics.

**Figure supplement 5.** Different estimates of initial resistant fraction lead to similar qualitative dynamics.

evolutionary dynamics not captured by the model. Intuitively, the experimental results can be explained by the strongly antagonistic interaction between the two drugs, which reverses the selection for resistance to CIP at sufficiently high TGC concentrations (compare CIP resistance at TGC = 0 and TGC = 0.04 g/mL, which both involve CIP ≈ 0.2 g/mL; similar results have been seen in other species and for other drug combinations; *Chait et al., 2007*). We also compared model predictions with experimental adaptation to two additional drug pairs (*Figure 5—figure supplements 1* and *2*) and again found that the model captures the qualitative features of both resistance changes to the component drugs and growth adaptation.

## Effects of mutations

While we have focused on selection dynamics, the model can be extended to include mutation events linking different phenotypes (i.e., linking different pairs of scaling parameters). These mutational pathways may be necessary to capture certain evolutionary features – for example, historical contingencies between sequentially acquired mutations (*Barbosa et al., 2017*; *Card et al., 2019*; *Das et al., 2020*). To incorporate mutational structure and processes, we modify the equation for each subpopulation $n_i$ to

$$\frac{dn_i}{dt} = g_i n_i - \mu n_i + \mu \sum_j^M m_{ji} n_j \tag{12}$$

where μ is the mutation rate and $m_{ji}$ is the probability that strain $j$ mutates to strain $i$, given that there is a mutation in strain $j$. We will refer to the matrix formed by the parameters $m_{ji}$ as the mutation matrix as it contains information about allowed mutational trajectories linking different phenotypes (*Day and Gandon, 2006*). The Price equation for the mean traits (the mean scaling parameters) now becomes

$$\frac{d\bar{\alpha}}{dt} = \text{Cov}_x(\alpha, g) - \mu(\bar{\alpha} - \bar{\alpha}_m),$$
$$\frac{d\bar{\beta}}{dt} = \text{Cov}_x(\beta, g) - \mu(\bar{\beta} - \bar{\beta}_m), \tag{13}$$

where $\bar{\alpha}_m$ and $\bar{\beta}_m$ are the mean trait values in all mutants that arise, and are given by

$$\bar{\alpha}_m = \sum_{ij} \alpha_i m_{ji} f_j,$$
$$\bar{\beta}_m = \sum_{ij} \beta_i m_{ji} f_j. \tag{14}$$

When the mutation matrix has a particularly simple structure, the effects of mutation will be similar to those of selection. For example, when mutations occur with uniform probability between the ancestral strain and each mutant, the evolutionary dynamics are qualitatively similar to those in the absence of mutation (*Figure 6—figure supplement 1*, compare to *Figure 4*). On the other hand, certain mutational structures can lead to new behavior. As an example, we consider a toy model consisting of four phenotypes defined by the level of resistance (sensitive, S, or resistant, R) to one

of two drugs. The phenotypes are designated SS $(\alpha = 1, \beta = 1)$, RS $(\alpha = 1/5, \beta = 1)$, SR $(\alpha = 1, \beta = 1/5)$, or RR $(\alpha = 1/5, \beta = 1/5)$. Note that these phenotypes do not assume, a priori, any particular relationships between the underlying genotypes. For example, each phenotype could correspond to a single mutation – in which case the RR phenotype would be said to exhibit strong cross-resistance – or alternatively, the phenotypes could correspond to single drug mutants (SR and RS) and double mutants (RR), which implies a particular sequence in which the mutations can be acquired. These two situations would lead to significant differences in the expected structure of the mutational matrix and, as we will see, in the evolutionary dynamics.

For simplicity, we consider two different mutational structures: one corresponding to direct and uniform pathways between the ancestor (SS) and all mutant phenotypes (SR, RS, and RR), and the second corresponding to sequential pathways from ancestor to single mutants (SR, RS) and then from single mutants to double mutants (*Figure 6*). To illustrate the impact of mutational structure, we consider two drugs that interact antagonistically and choose the external drug concentrations such that $D_1 > D_2$ (*Figure 6C*). While both mutational structures lead eventually to a population of double-resistant phenotypes (RR), the sequential pathway leads to slower adaptation of growth as the population evolves first toward the most fit single mutant (in this case, RS because the drug combination contains a higher concentration of drug 1) before eventually arriving at RR (*Figure 6C–E*). The specific trajectory will of course depend on both the drug interaction (the shape of the growth contours) and the specific dosage combination (see *Figure 6—figure supplement 2*).

There are other types of mutational constraints that can be implemented in the formalism, including explicit functional dependencies between different antibiotic resistance phenotypes. As an example, we assumed a distance-based mutation matrix where the mutation probability between different strains depends on the Euclidean distance between their scaling parameter pairs $(\alpha_i, \beta_i)$ according to $m_{ji} \sim e^{-d_{ji}}$, where $d_{ji} = \sqrt{(\alpha_i - \alpha_j)^2 + (\beta_i - \beta_j)^2}$. Intuitively, this structure means that mutations are more likely between strains with similar rescaling parameters (and therefore similar levels of resistance to the component drugs). As expected, this constrained mutational structure leads to slower growth rate adaptation than a model with a simple uniform mutation model (*Figure 6—figure supplement 3*). These effects are further compounded by statistical properties of the collateral structures (e.g., positive or negative correlations between the possible αs and βs; *Figure 3*), the specific (stochastic) realization of those scaling parameters (*Figure 6—figure supplement 4*), and the precise shape of two-drug growth surface. Note, however, that these dynamics depend fundamentally on the global mutation rate μ, which modulates the relative balance between selection and mutation in a given environment.

## Evolutionary dynamics under temporal sequences of drug combinations

Evolutionary dynamics in the presence of multiple drugs can be extremely complex. While past work has focused primarily on the use of either temporal sequences or (simultaneous) combinations of antibiotics, more complex scenarios are possible, in principle, but difficult to analyze even in theoretical models. The simplicity of the Price equation framework allows us to investigate evolutionary dynamics in response to a more complex scenario: temporal *sequences* of antibiotic *combinations*. As proof of principle, we numerically studied time-dependent therapies consisting of two sequential regimes (treatment A and treatment B; *Figure 7*) characterized by different dosage combinations of tigecycline (drug 1) and ciprofloxacin (drug 2) (the growth surface was measured in *Dean et al., 2020*). The two dosage combinations were chosen to lie along a contour of constant growth (*Figure 7A*), meaning that the net inhibitory effects of A and B are the same when applied to ancestral cells. Using experimental estimates for $\alpha, \beta$ (*Figure 3—figure supplement 1C*), we find that both the resistance levels to the two drugs and the growth rate increase during treatment, as one might expect. However, the dynamics of these changes depend on both the relative duration of each treatment and total treatment length (*Figure 7—figure supplements 1* and *2*). For example, consider a treatment of total length $T$ consisting of an initial period of treatment A followed by a final period of treatment B. If we vary the relative length of the two epochs while keeping the total treatment length fixed, we find that the resistance to each drug and the growth rate increase monotonically as the fraction of time in B increases (*Figure 7H–J*). This is a consequence of the interplay between the distribution of available mutants and the shape of the growth surface under these particular two drugs; the mutants tend to feature higher levels of resistance to drug 2 than drug 1, yet

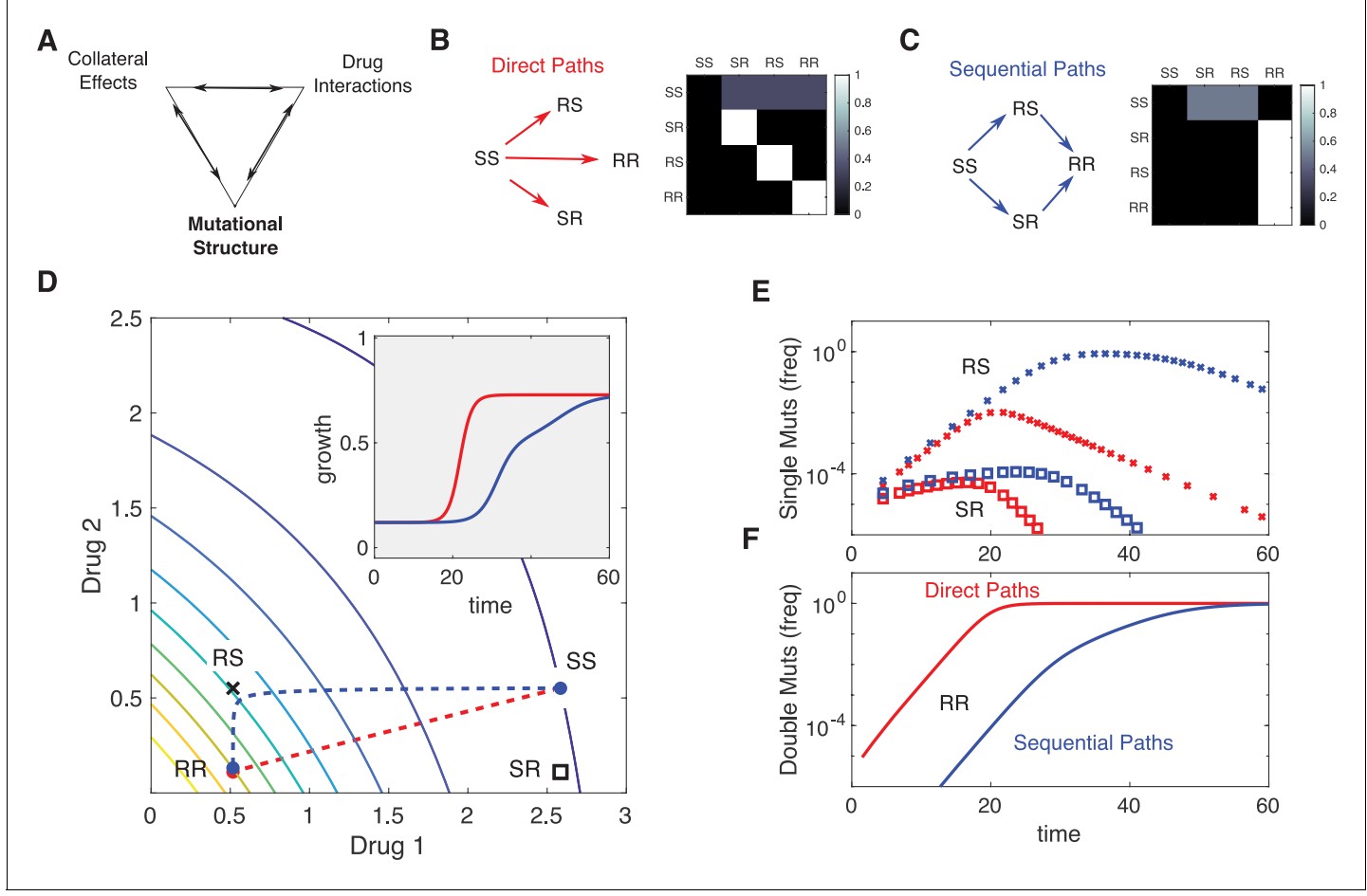

**Figure 6.** Incorporating mutational structure modulates evolutionary dynamics. (A) Evolutionary dynamics reflect a combination of collateral effects, drug interactions, and mutational structure. (B) Phenotypes are defined by relative resistance (R) or sensitivity (S) to each of two drugs, leading to four possible phenotypes: SS ($\alpha = 1, \beta = 1$), RS ($\alpha = 1/5, \beta = 1$), SR ($\alpha = 1, \beta = 1/5$), or RR ($\alpha = 1/5, \beta = 1/5$). Mutations can occur via direct paths (B), where each mutant phenotype is accessible directly from the fully sensitive ancestor, or via sequential paths (C), where the doubly resistant mutant (RR) is only accessible from single mutants (RS or SR). Entries of the mutation matrix $m_{ji}$ (heatmaps) reflect the probability of mutating from state $j$ (rows) to state $i$ (columns) given that a mutation occurs in state $j$. (D) Main panel: contour plot of growth surface in ancestral (SS) cells. For a specific choice of drug dosages, SS cells experience the true dosage combination ($D_1 \approx 2.6, D_2 \approx 0.6$; blue circle, SS) while single mutants (black x, RS; and black square, SR) and double-resistant mutants (RR) experience decreased effect concentration of one or more drugs. Dashed lines show mean trajectories ($\bar{\alpha}(t), \bar{\beta}(t)$) for direct (red) and sequential (blue) mutational structures. Inset: mean growth rate for direct (red) and sequential (blue) mutational structures. (E, F) Population frequencies for single mutants (panel D; xs are RS, squares are SR) and double mutants (panel E) under direct (red) or sequential (blue) mutational structures. See also *Figure 6—figure supplements 1–4*.

The online version of this article includes the following figure supplement(s) for figure 6:

**Figure supplement 1.** Adaptation in the presence of mutation depends on drug interaction and collateral effects of resistance.
**Figure supplement 2.** Mutational structure modulates evolutionary dynamics for additive and synergistic drug combinations.
**Figure supplement 3.** Uniform vs. distance-based mutation and evolutionary dynamics for 4 collateral structures.
**Figure supplement 4.** Stochastic realizations of collateral structures and evolutionary dynamics under two mutation models.

the benefits of this resistance – that is, the increase in growth rate due to rescaling the concentration of drug 2 – are favored much more so under condition B than condition A (where rescaling tends to produce effective drug concentrations that lie near the original growth contour). It is notable, however, that effects (both resistance levels and final growth) change nonlinearly as the fraction of time in B is increased – that is, even in this simple model, the effects of a two-epoch (A then B) treatment cannot be inferred as a simple linear interpolation between the effects of A-only and B-only treatments.

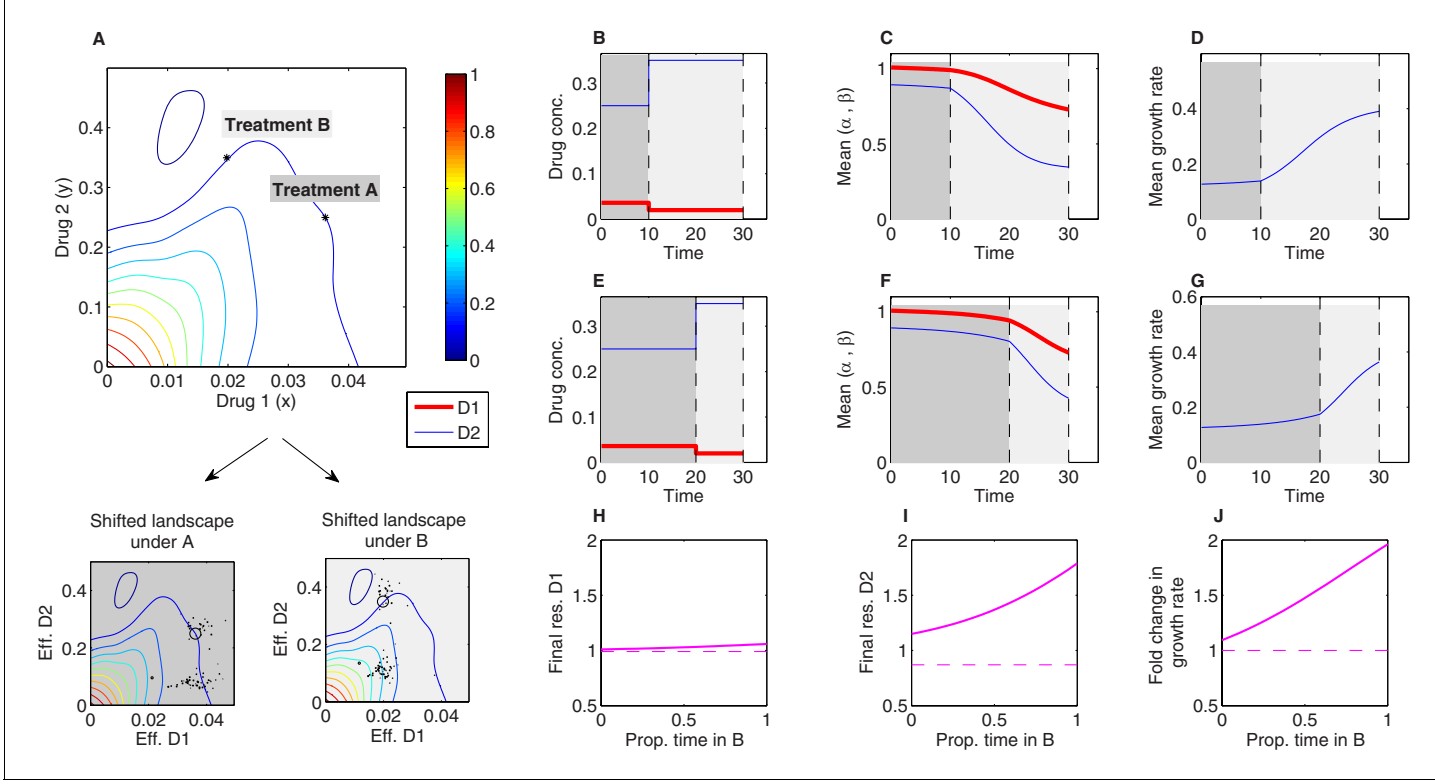

**Figure 7.** Sequential multidrug treatments lead to different evolutionary outcomes under different schedules. The model framework can be used to predict phenotypic evolution and growth dynamics in a bacterial population under consecutive treatments involving different dosage combinations of the same two drugs. (A) Treatments A and B (asterisks) are chosen to lie along a contour of constant growth in the space of drug concentrations. Two-drug growth surface (top panel) and the collection of scaling parameter pairs are based on data from *Dean et al., 2020* for tigecycline (drug 1) and ciprofloxacin (drug 2). The same set of scaling parameters leads to different effective drug concentrations when exposed to treatment A (bottom-left panel; each mutant is represented by a single point representing the effective concentration of each drug) and treatment B (bottom-right panel). (B-G) Time series of drug concentrations (B, E), mean scaling parameters (C, F), and mean growth rate (D, G) for sequential treatments (treatment A followed by treatment B). Panels (B–D) correspond to a short epoch (10 time units) of A followed by a longer (20 time units) epoch of B. Panels (E–G) correspond to a long epoch of A followed by a shorter epoch of B. (H–J) Changes in resistance to drug 1 ($1/\bar{\alpha}$) (H), resistance to drug 2 ($1/\bar{\beta}$) (I), and growth rate (J) for treatments of a fixed length ($T = 10$ time units) but varying the fraction of time spent in each treatment. Each treatment consisting of one epoch of A (for time $t_1$) followed by one epoch of B (for time $T - t_1$). See also *Figure 7—figure supplement 1* (for detailed temporal trajectories) and *Figure 7— figure supplement 2* (for longer total therapies with saturating dynamics). In all panels, we assumed mutants were characterized by scaling parameter pairs ($\alpha_i$ and $\beta_i$) measured experimentally (*Figure 3—figure supplement 1*). Initial populations consisted of primarily ancestral cells (fraction 0.99), while the remaining population (fraction 0.01) was uniformly distributed among all available mutants.

The online version of this article includes the following figure supplement(s) for figure 7:

**Figure supplement 1.** Detailed temporal trajectories of population mean traits and growth rate under sequential two-drug therapies.

**Figure supplement 2.** Detailed temporal trajectories of population mean traits and growth rate under sequential two-drug therapies for longer total duration.

## Discussion

Antibiotic resistance is a growing threat to modern medicine and public health. Multidrug therapies are a promising approach to both increase efficacy of treatment and prevent evolution of resistance, but the two effects can be coupled in counterintuitive ways through the drug interactions and collateral effects linking the component drugs. Our results provide a unified framework for incorporating both drug interactions and collateral effects to predict phenotypic adaptation on coarse-grained fitness landscapes that are measurable features of the ancestral population (*Table 1*). Special cases of the model reproduce previous experimental results that appear, on the surface, to be contradictory; indeed, adaptation can be driven primarily by drug interactions, primarily by collateral effects, or by a combination of both, and the balance of these effects can be shifted using tunable properties of the system (i.e., the ratio of drug dosages). Our model was inspired by rescaling arguments that

**Table 1.** Different biological properties correspond to different features of the model.

| Biological property | Model property |
| --- | --- |
| Scaling parameters (ancestral) | $\alpha = \beta = 1$ |
| Resistance to drug 1* | Inverse scaling parameter $\alpha^{-1}$ |
| Resistance to drug 2* | Inverse scaling parameter $\beta^{-1}$ |
| Synergistic interaction | Locally convex isoboles in 2-drug growth surface |
| Antagonistic interaction | Locally concave isoboles in 2-drug growth surface |
| Cross-resistance | Positively correlated scaling parameters $(\alpha, \beta)$ |
| Collateral sensitivity | Negatively correlated scaling parameters $(\alpha, \beta)$ |
| External concentration $(x, y)$ | Effective drug concentration $(\alpha x, \beta y)$ |
| Fitness cost | Prefactor $g \rightarrow (1 - \gamma)g$ |
| Resistance-dependent fitness cost | $\gamma \equiv \gamma(\alpha, \beta)$ |
| De novo mutation | Mutation matrix $m_{ji}$ |

*Fold change in minimum inhibitory concentration or similar inhibitory concentration.

were originally introduced in *Chait et al., 2007* and have since been shown to capture phenotypic features of dose-response surfaces or adaptation in multiple bacterial species (*Michel et al., 2008*; *Hegreness et al., 2008*; *Torella et al., 2010*; *Wood and Cluzel, 2012*; *Wood et al., 2014*; *Dean et al., 2020*; *Das et al., 2020*). Our results complement these studies by showing how similar rescaling assumptions, when formalized in a population dynamics model, lead to testable predictions for the dynamics of both growth adaptation and phenotype (resistance) evolution. Importantly, the model also has a simple intuitive explanation, with evolutionary trajectories driven by weighted gradient dynamics on two-dimensional landscapes, where the local geometry of the landscape reflects the drug interaction and collateral effects constrain the direction of motion. We have also illustrated how de novo mutation can be integrated in the same framework, provided the mutational structure among a pool of possible phenotypes is known or if specific assumptions are made regarding functional links and constraints for shifts between different resistance levels. Mutation at constant rate is indeed a classic version of the complete Price equation. However, more complex mutational effects – including stress-dependent modulation of mutation rate (*Kohanski et al., 2010*; *Vasse et al., 2020*) – could be included as a more flexible tunable term in *Equation 5*.

It is important to keep in mind several limitations of our approach. The primary rescaling assumption of the model is that growth of a drug-resistant mutant is well approximated by growth of the ancestral strain at a new 'effective' drug concentration – one that differs from the true external concentration. This approximation has considerable empirical support (*Chait et al., 2007*; *Wood and Cluzel, 2012*; *Wood et al., 2014*; *Das et al., 2020*) but is not expected to always hold; indeed, there are examples where mutations lead to more complex nonlinear transformations of the drug-response surface (*Wood et al., 2014*; *Munck et al., 2014*). In addition, it is possible that selection can act on some other feature of the dose-response curve characterizing single-drug effects – modulating, for example, its steepness (rather than merely its scale). While these effects could in principle be incorporated into our model – for example, by assuming transformations of the ancestral surface, perhaps occurring on a slower timescale, that go beyond simple rescaling – we have not focused on those cases. For simplicity, we have also neglected a number of features that may impact microbial evolution. For example, we have assumed that different subpopulations grow exponentially, neglecting potential interactions including clonal interference (*Gerrish and Lenski, 1998*), intercellular (*Koch et al., 2014*; *Hansen et al., 2017*; *Hansen et al., 2020*) or intra-lineage (*Ogbunugafor and Eppstein, 2017*) competition, and cooperation (*Yurtsev et al., 2013*; *Sorg et al., 2016*; *Estrela and Brown, 2018*; *Frost et al., 2018*; *Hallinen et al., 2020*), as well as potential effects of demographic noise and population extinction (*Coates et al., 2018*). These complexities could also be incorporated in our model, perhaps at the expense of some intuitive interpretations (e.g., weighted gradient dynamics) that currently apply. In addition, we have not explicitly included a fitness cost of resistance (*Andersson and Hughes, 2010*) – that is, we assume that growth rates of mutants and ancestral cells are identical in the absence of drug. This assumption could be relaxed by including a prefactor to

the growth function, $g_i \to (1 - \gamma_i(\alpha_i, \beta_i))g_i$, where $\gamma_i(\alpha_i, \beta_i)$ is the cost of resistance, which in general depends on the scaling parameters (if not, it can be easily incorporated as a constant). While such fitness costs have traditionally been seen as essential for reversing resistance (with, e.g., drug-free 'holidays'; *Dunai et al., 2019*), our results underscore the idea that reversing adaptation relies on differential fitness along a multidimensional continuum of environments, not merely binary (drug/no drug) conditions. Our results indicate resistance and bacterial growth can be significantly constrained by optimal tuning of multidrug environments, even in the absence of fitness cost. Finally, our model deals only with heritable resistance and therefore may not capture phenotypic affects associated with, for example, transient resistance (*El Meouche and Dunlop, 2018*) or cellular hysteresis (*Roemhild et al., 2018*).

Our goal was to strike a balance between analytical tractability and generality vs. biological realism and system specificity. But we stress that the predictions of this model do not come for free; they depend, for example, on properties of the dose-response surfaces, the collection of scaling parameters, and the specific mutational structure. In many cases, these features can be determined empirically; in other cases, some or all of these properties may be unknown. The evolutionary predictions of the model will ultimately depend on these inputs, and it is difficult to draw general conclusions (e.g., 'synergistic combinations always accelerate resistance') that apply across all classes of drug combinations, collateral effects, and mutational structures. But we believe the framework is valuable precisely because it generates testable predictions in situations that might otherwise seem intractably complex.

Our approach also has pedagogical value as it connects evolutionary effects of drug combinations and collateral effects with well-established concepts in evolutionary biology and population genetics (*Price, 1970*; *Price, 1972*; *Day and Gandon, 2006*). Approximations similar to *Equation 9* have been derived previously in quantitative genetics models (*Abrams et al., 1993*; *Taylor, 1996*) and other applications of the Price equation (*Lehtonen, 2018*). While the gradient approximation does not require that the population be monomorphic with rare mutants or a particular form for the phenotype distribution, it does require that the majority of trait values (here scaling parameters) are contained in a regime where $G(x, y)$ is approximately linear; in our case, that linearity arises by Taylor expansion and neglecting higher-order deviations from the population mean. More generally, the direction of evolutionary change in our model is determined by the gradient of the fitness function $\nabla G$ evaluated at the mean trait $(\bar{\alpha}, \bar{\beta})$; when the gradient vanishes, this point corresponds to a singular point (*Waxman and Gavrilets, 2005*) or an evolutionarily singular strategy (*Geritz et al., 1998*). Whether such point may be reached (convergence stability) and how much variance in trait values can be maintained around such point (whether evolutionarily stable [ESS]) depend on other features, such as higher-order derivatives (*Otto and Day, 2011*; *Eshel et al., 1997*; *Lehtonen, 2018*; *Smith, 1982*; *Parker and Smith, 1990*). In the case of drug interactions, the fitness landscape in $(\alpha, \beta)$ space will typically have a single maximum at (0, 0) corresponding to effective drug concentrations of zero. However, whether that point is reachable in general or within a given time frame will depend on the available mutants preexisting at very low frequencies in the population, or on the speed and biases in the mutational process itself, if such mutants are to be generated de novo during treatment. In principle, the model also allows for long-term coexistence between different strains; in that case, the rescaled effective drug concentrations experienced by both strains would fall along a single contour of constant growth. Hence, while variance in the population growth will necessarily decrease over time, variance in the traits (scaling parameters) themselves can change non-monotonically.

The framework is sufficiently flexible to integrate different strands of empirical data, and our results underscore the need for quantitative phenotype data tracking resistance to multiple drugs simultaneously, especially when drug combinations are potentially driving selection dynamics. At an epidemiological level, the dominant approach in describing resistance has been to use fixed breakpoints in MIC and track the percentage of isolates with MIC above (drug-resistant) or below (drug-sensitive) such breakpoint (e.g., *ECDC, 2019*; *Chang et al., 2015*). By missing or decoupling patterns of co-occurrence between MICs to different drugs across isolates, this approach remains incomplete for mechanistic predictions. Our framework suggests that going beyond such binary description towards a more continuous and multidimensional phenotype characterization of drug resistance is possible, with applications not just in microbiology but also in the evolutionary epidemiology of drug resistance (*Day and Gandon, 2012*; *Day et al., 2020*). In the long run, these advances

may yield better and more precise predictions of resistance evolution at multiple scales, and, in turn, optimized treatments that balance short-term inhibitory effects of a drug cocktail with its inseparable, longer-term evolutionary consequences.

Perhaps most importantly, our approach provides a low-dimensional approximation to the high-dimensional dynamics governing the evolution of resistance. In contrast to the classical genotype-centric approaches to resistance, our model uses rescaling arguments to connect measurable traits of resistant cells (scaling parameters) to environment-dependent phenotypes (growth). This rescaling dramatically reduces the complexity of the problem as the two-drug-response surfaces – and, effectively, the fitness landscape – can be estimated from only single-drug dose-response curves. Such coarse-grained models can help extract simplifying principles from otherwise intractable complexity (*Shoval et al., 2012*; *Hart et al., 2015*). In many ways, the classical Price equation performs a similar function, revealing links between trait-fitness covariance and selection that, at a mathematical level, are already embedded in simple models of population growth. In the case of multidrug resistance, this formalism reveals that drug interactions and collateral effects are not independent features of resistance evolution, and neither, alone, can provide a complete picture. Instead, they are coupled through local geometry of the two-drug-response surface, and we show how specific dosage combinations can shift the weighting of these two effects, providing a framework for systematic optimization of time-dependent multidrug therapies.

## Materials and methods

### Estimating scaling parameters from experimental dose-response curves

The scaling parameters for a given mutant can be directly estimated by comparing single-drug dose-response curves of the mutant and ancestral populations. To do so, we estimate the half-maximal inhibitory concentration ($K_i$) for each population by fitting the normalized dose-response curve to a Hill-like function $g_i(d) = (1 + (d/K_i)^h)^{-1}$, with $g_i(d)$ the relative growth at concentration $d$ and $h$ a Hill coefficient measuring the steepness of the dose-response curve using nonlinear least-squares fitting. The scaling parameter for each drug is then estimated as the ratio of the $K_i$ parameters for the ancestral and mutant populations. For example, an increase in resistance corresponds to an increase in $K_i$ for the mutant population relative to that of the ancestor, yielding a scaling parameter of less than 1. Estimates for the scaling parameters for the three drug combinations used here are shown in *Figure 3—figure supplement 1* (from data in *Dean et al., 2020*).

While it is straightforward to estimate the scaling parameters for any particular isolate, it is not clear a priori which isolates are present at time 0 of any given evolution experiment. To compare predictions of our model with lab evolution experiments, we first estimated scaling parameters for all isolates collected during lab evolution experiments in each drug pair (*Dean et al., 2020*). This ensemble includes 50–100 isolates per drug combination and includes isolates collected at different timepoints during the evolution (after days 1, 2, or 3) as well as isolates selected in different dosage combinations *Figure 3—figure supplement 1*. We then randomly sampled from this ensemble to generate low-level standing diversity (on average, approximately 10 distinct pairs of scaling parameters) at time 0 for each simulation of the model, and we repeated this subsampling 100 times to generate an ensemble of evolutionary trajectories for each condition.

The results of the simulation can, in principle, depend on how these scaling parameters are sampled. While the qualitative differences between simulations do not depend heavily on this choice of subsampling in the data used here (*Figure 5—figure supplement 3*), one can imagine scenarios where details of the subsampling significantly impact the outcome. Similarly, precise comparison with experiment requires accurate estimates for the total evolutionary time and for the initial frequency of all resistant mutants, though for these data the qualitative results do not depend sensitively on these choices (*Figure 5—figure supplement 4* and *Figure 5—figure supplement 5*). We stress that these are not fundamental limitations of the model itself, but instead arise because we do not have a precise measure of the standing variation characterizing these particular experiments. In principle, a more accurate ensemble of scaling parameters could be inferred from cleverly designed fluctuation tests (*Luria and Delbrück, 1943*) or, ideally, from high-throughput, single-cell phenotypic studies (*Baltekin et al., 2017*). At a theoretical level, subsampling could also be modulated to

simulate the effects of different effective population sizes, with standing diversity expected to be significantly larger for large populations.

## Acknowledgements

This work was supported by Fundação Luso-Americana para o Desenvolvimento (FLAD/NSF grant-274/2016 to EG) and Instituto Gulbenkian de Ciência, the National Science Foundation (NSF No. 1553028 to KW), and the National Institutes of Health (NIH No. 1R35GM124875 to KW). The Center for Stochastic and Computational Mathematics is supported by FCT via UIDB/04621/2020 and UIDP/04621/2020.

## Additional information

### Funding

| Funder | Grant reference number | Author |
| --- | --- | --- |
| National Institutes of Health | 1R35GM124875 | Kevin B Wood |
| National Science Foundation | 1553028 | Kevin B Wood |
| Fundação Luso-Americana para o Desenvolvimento | 274/2016 | Erida Gjini |
| Calouste Gulbenkian Foundation | | Erida Gjini |
| FCT | UIDB/04621/2020 | Erida Gjini |
| FCT | UIDP/04621/2020 | Erida Gjini |

The funders had no role in study design, data collection and interpretation, or the decision to submit the work for publication.

### Author contributions

Erida Gjini, Kevin B Wood, Conceptualization, Formal analysis, Investigation, Methodology, Writing - original draft, Writing - review and editing

### Author ORCIDs

Erida Gjini https://orcid.org/0000-0002-0493-3102
Kevin B Wood https://orcid.org/0000-0002-0985-7401

### Decision letter and Author response

Decision letter https://doi.org/10.7554/eLife.64851.sa1
Author response https://doi.org/10.7554/eLife.64851.sa2

## Additional files

### Supplementary files

- Transparent reporting form

### Data availability

Data used in this paper was taken from a public repository: Dean, Ziah; Maltas, Jeff; Wood, Kevin (2020), Antibiotic interactions shape short-term evolution of resistance in *Enterococcus faecalis*, Dryad, Dataset, https://doi.org/10.5061/dryad.j3tx95x92 There are no restrictions on any new results.

The following previously published dataset was used:

| Author(s) | Year | Dataset title | Dataset URL | Database and Identifier |
|---|---|---|---|---|
| Ziah D, Maltas J, Wood K | 2020 | Data from: Antibiotic interactions shape short-term evolution of resistance in *Enterococcus faecalis* | https://doi.org/10.5061/dryad.j3tx95x92 | Dryad Digital Repository, 10.5061/dryad.j3tx95x92 |

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
