## [Decision Letter]

**Acceptance summary:**

Your manuscript offered a provocative integration of the Price equation with fundamental theory in population genetics, in the context of a relevant biomedical problem. We found the approach creative and rigorous, and is the type of cross-disciplinary theoretical work that evolutionary medicine could benefit from.

**Decision letter after peer review:**

Thank you for submitting your article "Price equation captures the role of drug interactions and collateral effects in the evolution of multidrug resistance" for consideration by *eLife*. Your article has been reviewed by 3 peer reviewers, one fo whom is a member of our Board of Reviewing Editors, and the evaluation has been overseen by George Perry as the Senior Editor. The reviewers have opted to remain anonymous.

Summary:

I thank the authors for submitting a compelling and well-executed study. The reviewers and reviewing editor agree that the manuscript offers an interesting take, is technically sound and provides a provocative synergy between modern questions in the evolution of antibiotic resistance, and classical theory in the form of the Price equation.

Some of the discussion between the reviewers involved the question of whether the paper's main argument, as outlined in the draft, was too strongly bound to the usage of the Price equation. That is, that the main result can be couched in terms of the Price equation may or may not mean that the finding is a true contribution to modern examinations of the evolution of antibiotic resistance, a field that is increasingly sophisticated in its use of both theory and technology.

I especially applaud the author's transparency: the issue is not that they oversold their result. To the contrary-the limitations were responsibly articulated. A lingering question is whether the result, as reported, is anchored to a slice of reality that is too thin to be broadly useful. This is an admittedly subjective area of discussion, but was a theme from the reviews and discussions.

In order to assuage some of the concerns, I have provided a list of recommended changes, collated from the formal; reviews and further exchanges with the reviewers, that could make the manuscript stronger.

I thank the authors for their hard work. I want to especially acknowledge that we are all working in times that are challenging for many reasons. I congratulate the authors on producing a strong piece of science under the circumstances.

Essential Revisions:

1) Please propose some treatment of the setting in the result that includes mutation and other stochastic processes. Several reviewers made mention of the "model" that was used. Some had specific issues with the notion that the rescaling in the form of the Price was truly a model at all. Others had the related concern that the paper, as it stands, applies to the very strict circumstance: that of a small population of equal numbers of every potential mutant in the model. For example, the authors state:

"We assume a finite set of M subpopulations (mutants), (i = 1…M), with each subpopulation corresponding to a single pair of scaling parameters. For simplicity, we assume each of these mutants is initially present in the population at low frequency and neglect mutations that could give rise to new phenotypes, though it is straightforward to incorporate them into the same framework"

If it is indeed "straightforward," then it would be great to see this explored, with new mutations and other stochastic processes. We recognize that the strength of the study is in its elegance re-scaling, however, there are concerns that the use case is small enough that it limits its applicability.

2) Somewhat relatedly, the reviewers had some issues with use of the term "model" to describe the Price equation. While the mathematics and framing themselves are quite responsible, there were some particular issues raised about the framing. I want to especially point you towards the comments from reviewer #2, who explained issues with usage of the term "model" and "prediction" throughout.

3) There are several issues with figures, figure labels, captions and descriptions that require the author's attention. As this paper does hinge on an understanding of the particulars, it is especially important that figures clearly communicate the results. Please see the individual reviewer's comments for specifics.*Reviewer #1:*

This study utilizes the price equation in order to describe evolution of drug resistance, recognizing the cutting-edge problem of predicting the evolution of resistance in light of collateral effects and drug-drug interactions.

The manuscript offers a formalism that cleverly integrates theory across paradigms, from quantitative genetics to the evolution of antibiotic resistance. As an intellectual exercise, this is courageous, the type of integrative thinking necessary in the modern iteration of evolutionary medicine.

I am not, however, of the mindset that the Price equation is so important a framing that its invocation/application to a different paradigm (antibiotic resistance) is especially interesting by itself. Said differently, I am not personally intrigued by the a study that is successful at connecting antibiotic resistance to the Price equation. Alternatively, I am much much more intrigued by the possibility that the Price equation might add a richer and more nuanced overall perspective to cutting-edge problems in the evolution of antibiotic resistance.

And I can say that, on the balance, the manuscript was successful in offering a new take on relevant problem.

Also, this study carries out its methods carefully, and its formalisms are well-written. The essential idea is that the vagaries of drug interactions and collateral effects can be collapsed into a formalism that considers both when predicting the trajectory of drug resistance, is a fascinating and important one.

The study is especially transparent (refreshingly so) about the limitations of its scope. And while this transparency is a legitimate positive of this story, I did find myself concerned about the small number of cases that theory proposed in this study applies to. For example, the Price equation as applied in the study applies to selection on standing variation, and not for the step-wise acquisition of resistance that fitness landscapes are often constructed to study. The authors argue that this type of nuance can be engineered into the formalism, which is not obvious to me (I'm not suggesting that it isn't true, only that it isn't obvious).

But that this study has a relatively thin application space does not make the study irrelevant. For the outlined problem space, the arguments appear sound.

I must say that given the current state of the problem of antimicrobial resistance in 2021, sometimes featuring complex, highly dynamic, epistastic fitness seascapes, for example, I was hoping for framework that could be used for fitness landscapes as they are currently constructed and studied.

On the balance, however, I found the study to be well-constructed, written, and the formalisms to be correct. The main concerns regarding the scope might be a matter of preference. And even I must recognize that responsible theoretical biology often requires a narrow scope in order to generate formalisms. The idea here is that from this contribution, the community can then extend the study to other such systems and data sets.

I would like the authors to consider, and possibly cite, if they are relevant, more recent papers by Pamela Yeh on this topic:

Tekin, Elif, Pamela J. Yeh, and Van M. Savage. "General form for interaction measures and framework for deriving higher-order emergent effects." Frontiers in Ecology and Evolution 6 (2018): 166.

Tekin, Elif, Van M. Savage, and Pamela J. Yeh. "Measuring higher-order drug interactions: A review of recent approaches." Current Opinion in Systems Biology 4 (2017): 16-23.

Tekin, Elif, et al., "Enhanced identification of synergistic and antagonistic emergent interactions among three or more drugs." Journal of The Royal Society Interface 13.119 (2016): 20160332.

There are some small issues with the figures that need to be resolved:

I found Figure 6 to be complicated. I would suggest the authors re-generate it with the goal of explaining it to someone not well-versed in the system.

I had some issues with the legends in Figures 4 and 5. Ensure that they describe the information in the figures properly.*Reviewer #2:*

This paper examines the interaction between pharmacological and evolutionary factors that mediate the emergence of resistance to combination antimicrobial therapy. The authors introduce and demonstrate the utility of a mathematical equation that describes how both (a) the pharmacological interaction between two drugs (e.g. synergy vs antagonism) and (b) the interaction between the fitness effects of mutations that confer resistance to one drug or the other (or both), jointly determine the direction of selection in the presence of two-drug therapy. They show that the equation they describe follows the same form as a two-trait version of the classical "Price equation" in population genetics. They also examine a "weak selection" limit (when evolution proceeds by selection of a sequence of strains each with only a slight change in resistance level from the previous). In this limit their equation has an even more intuitive physical meaning as movement along an weighted version of the gradient of the drug's dose-response curve, where the weighting reflects the nature of the cross-resistance profile. Many examples of the implications of this equation for different drug interactions and different fitness landscapes are presented. The results of this paper unite findings from quite a few previous studies in a common framework. For example, the authors show how to understand (i) when resistance to one drug in a combination will be more strongly selected than resistance to the other and how this direction of selection can change based on the relative levels of two drugs in a combination – even if total inhibition is the same, (ii) when drug antagonism (vs synergy) is morel likely to slow down the rate of adaptation, and (iii) how the relative ordering and frequency of cycling between two drugs in a combination impacts the rate of resistance emergence.

It is important for readers to understand that the Price equation – and thus by extension the result in this paper – is not really a mathematical "model" that includes assumptions about the underlying biological processes and makes predictions about potential experimental results. Instead, the Price equation is a mathematical truth that simply involves decomposing certain statistical relationships between variables and expressing them in an intuitive way. Roughly, in this case the Price equation describes how the average level of resistance in a population of microbes will change over time depending on the frequencies of different strains in the population and their resistance levels, including the correlation between resistance to each drug. Thus, this paper is primarily purely theoretical/mathematical work.

The authors do apply their equation to some real-life drug profiles and fitness landscapes using dose-response curves for two-drug antibiotic combinations to calculate the direction of selection for different dose combinations. Later in the paper, they do also apply their equation as a "model" to predict entire evolutionary trajectories, by additionally assuming that they can define a priori all the available resistance mutations and ignore any additionally generated mutations. Then, they show that the level of resistance selected after a specific time as predicted by their equations agrees relatively well with the results of in vitro experiments containing these same mutations. This is actually quite a surprising and nice result.

Overall this paper provides a useful new framework for understanding drug resistance evolution and provides multiple examples of insights offered by this framework. They provide a nice connection between the mainly mathematical nature of their results and scenarios involving antibiotic treatment and resistance evolution in the lab. The introduction of this paper is very comprehensive and summarizes a huge body of work on the evolution of drug resistance. The discussion is equally thorough and mentions most of the caveats I describe here.

Although the authors of the paper focus on bacterial infections and antibiotic resistance, I think the results are much more general, and likely also apply to multi-drug therapy to viral infections (e.g. HIV, HCV), parasitic infections (e.g. malaria), and even anti-cancer therapy.

While not a weakness that is specific to this paper, it is important for readers to understand that a Price equation only describes the act of selection acting on existing genetic variation in a population, and does not describe the act of generating that variation (i.e. mutations). Thus, while this approach can be helpful for understanding the direction and speed of selection in one vs two-drug combinations of different types, it cannot be used to understand the overall risk of resistance emergence, since the rate at which mutations are generated may also differ. For example, it often takes two separate mutations to generate resistance to two drugs with different mechanisms of action, so resistance to two-drug therapy is often slowed by the rate of generating resistant mutations. In addition, the rate at which mutations are generated during therapy depends on the population size over time, which might differ along different selective paths described by the Price equation for different drug combinations. Therefore, there are some important results from previous work related to factors that accelerate or inhibit multi-drug resistance which cannot be understood in the framework presented in this paper.

One assumption that the authors use in order to get a simplified Price-equation-like result is that the effect of a drug resistance mutation on the dose-dependent effect of a drug on microbial growth can be described by a single parameter, which just shifts the drug concentration to a lower value. This is equivalent to saying that a mutation simply alters the IC50 of the drug. However, many studies have shown that in general mutations have at least one and often two other effects on dose-response curves: firstly, they have a fitness cost, meaning that the maximum fitness in the absence of drug is lower, and secondly, they can also alter the slope of the dose response curve. It is unclear how these effects change the equation the authors have derived.

The impact of the authors approach on studying resistance evolution in any particular system depends on the regime that system is in. If resistance tends to evolve by selection slowly acting on an extremely diverse population containing strains with many different resistance levels, then this approach will have the most utility. Alternatively, if resistance evolution tends to be limited by rare mutational events that produce strains with large differences in resistance levels from the parental strain, then this approach will have less use.

To summarize, the most important caveat for readers to understand is that the mathematical expressions that come out of this paper should not be said to be able to "predict" evolution, since they say nothing about the availability of possible resistance mutations or the rate at which they could be generated.

Concerns and suggested changes/clarifications:

– It is a bit misleading to call the Price Equation a mathematical model. It's not a model in the way that word is generally used in biology – it doesn't descibe a mechanism or make any assumptions, and it's not subject to experimental verification. It's just a mathematical statement about a statistical relationship among variables, based on the definition of those variables and basic laws of calculus. The authors should be very clear about this throughout the paper.

– The parts of the paper that are more like a "model" are the parts where you use the Price Equation to describe an evolutionary trajectory, assuming a particular distribution of mutants from standing genetic variation and ignoring additional mutations, or, when you assume a weak selection regime.

– Overall the authors should be more careful about the use of the word "model" and claims about "predictions" throughout.

– I'm not sure about Equation 2 – 5 requires the assumption that the effect of a mutant can be described by a rescaling of drug concentrations. Instead, I think you could replace the α parameters with another parameter r for example that simply means "resistance level" but does not specify how the "resistance level" changes the shape of the dose-response curve.

– Equation 5 actually is only half of the regular Price equation. It is the "selection" component but is missing the "transmission" component. The authors should be clearer about what processes relevant to the evolution of drug resistance they are ignoring that allow them to set this second term to zero.

Comments on Figures and results:

Figure 2 – What are the units of the axes on all these graphs?

a) I don't understand this statement in the caption: "We then simulated an external condition where the combination drug concentration is held constant at 0.6 of the maximal concentration of drug 1 and 0.1 of the maximal concentration of drug 2.". There is no such thing as a "maximal" drug concentration … you can always keep adding more and more drug. It would be clearer just to state the drug level used in absolute terms. Or maybe you mean the "highest tested drug level"?

Figure 3 – Some of the results shown here are a little confusing and hard to understand. In the black scenario, there are few available resistant strains with lower α, so we expect selection to proceed in the direction of Drug 2/β.. that agrees with the results in B. But for the blue case where, despite lots of available α but little available β, the trajectory in B seems relatively symmetrical in both drugs … why? Also I was surprised that the case of collateral sensitivity (green) wasn't the worst case … but I realized it's a bit misleading because the four scenarios don't only differ in the nature of the correlations in resistance levels, but also in the magnitude of the variance in α or β and hence the best possible available mutation. I think that it would be a fairer comparison if for the green and magenta scenarios the variance was narrower around the lines of perfect correlation/anticorrelation.

Figure 4 – This is nice figure that really summarizes the combined effects of drug interactions and cross-resistance profile. However this figure also summarizes the major limitations of this approach. The early slopes of each growth rate vs time curve do show how these factors interact to determine selection strength. However, the fact that the maximum growth rate varies by curve is really just an artifact of the arbitrary and limited available mutants the authors have created for selection to act on. In reality, mutations are continually and sequentially generated over time, and this rate of generation of mutants also impacts the increase in growth rate over time and might remove any "maximum" resistance level that is achievable.

Figure 5 – Are these predicted trajectories using the weak selection limit or the full equations without this additional assumption?

– I'm assuming there was some fixed time limit that selection was allowed to proceed for in these experiments and that it was the same in all cases?

– B: is the y axis absolute IC50 or change in IC50? The axes titles just say IC50 but the figure caption says "relative change in IC50". Also it would be helpful if the baseline IC50 (ie for the wild type strain) where shown on these graph as well. Finally, I don't understand how the results for the TGC IC50 agree with the results in A. From B, the IC50 looks totally flat with TGC concentration. But from A, it looks like there is a large increase in TGC resistance (increase in TGC IC50) for high TGC concentrations, but little TGC resistance for low TGC concentrations.

– C: I don't understand what the y axis is here. If this is the increase in growth rate, then why is it highest for low TGC concentrations, yet both panels in B show no change in IC50 to either drug at low TGC concentrations? And any idea what could be causing the systematic difference between the predicted and observed changes in growth rate?

Figure 6 – This figure was quite confusing, it needs more annotation on the figure itself (not just written in the caption), and some panels are unnecessarily small (with unnecessary white space between them). For panels H and K, are there better axis limits that could be used so that 95% of the figure is not all one color? Also I found it is hard to see in this figure any of the differences between the outcomes of the two different orderings that are described in the text.

*Reviewer #3:*

In this work, Gjini and Wood create a model to simulate evolutionary dynamics of antibiotic resistance arising as a response to drug combinations. They demonstrate that evolution to these drug combinations is a variation of the Price equation and show that some evolutionary paths experimentally correspond to a weighted gradient descent on the growth landscape.

The work appears mathematically sound, though we have concerns about the scope and generality of the results. In particular this work appears to rely fundamentally on the assumption that drug resistance acts as a linear rescaling of the two-drug dose response curve. We believe this work would be strengthened by making it more clear what, beyond the mathematical description of the rescaling hypothesis they cite extensively, this work adds to the literature. Since the model appears to also require an a priori description of the resistance characteristics of every possible accessible mutant, which typically is not known, it isn't clear the utility of the model.

– This paper models evolution in the presence of physiological interactions between antibiotics but not evolutionary interactions. The fundamental assumption that evolution of resistance scales the surface but does not change its shape, which is central to the findings, is true in some cases, though violated for many mutations (the authors cite numerous papers on cross-resistance and collateral sensitivity that are examples of this).

– While the authors describe it as "straightforward" to include discrete mutational events instead of a small population of equal numbers of every potential mutant in the model, it is not clear that this is the case. For example, a population exposed to an above-MIC concentration of a bacteriostatic drug will no longer generate additional mutants, and so the standing diversity before drug application matters critically. As it stands the model is really about differential growth and not evolution per se, since neither mutation nor complete extinction can occur.

– It isn't clear the experiments are representative of nontrivial behavior predicted by the model. For example in figure 5 no adaptation to TGC was seen, so it seems the only real evolution was rescaling CIP, which wouldn't yield the suppression-induced selection against resistance they then discuss. These results could be entire an artifact of the selection being entirely on CIP and not at all on TGC. The evolution experiments in Figures S3 and S4 seem much more compelling, though.

– If it is indeed straightforward to add mutational events, this would make the model appreciably more realistic.

---

## [Author Response]

Essential Revisions:1) Please propose some treatment of the setting in the result that includes mutation and other stochastic processes. Several reviewers made mention of the "model" that was used. Some had specific issues with the notion that the rescaling in the form of the Price was truly a model at all. Others had the related concern that the paper, as it stands, applies to the very strict circumstance: that of a small population of equal numbers of every potential mutant in the model. For example, the authors state:"We assume a finite set of M subpopulations (mutants), (i = 1…M), with each subpopulation corresponding to a single pair of scaling parameters. For simplicity, we assume each of these mutants is initially present in the population at low frequency and neglect mutations that could give rise to new phenotypes, though it is straightforward to incorporate them into the same framework"If it is indeed "straightforward," then it would be great to see this explored, with new mutations and other stochastic processes. We recognize that the strength of the study is in its elegance re-scaling, however, there are concerns that the use case is small enough that it limits its applicability.

In the revised manuscript, we include a new section (including 1 additional figure, Figure 6, and several associated supplemental figures) that incorporates mutation. Under these conditions, the Price Equation includes an additional term that depends on a mutation matrix that describes the probability of mutation between any pairs of strains (Equation 13). Under some conditions (e.g. when mutations occur with uniform probability between the ancestor cells and each mutant), the effects of mutation are qualitatively similar to those of selection (Figure 4 and Figure 6 figure supplement 1). On the other hand, mutational matrices can be chosen to reflect known features of the mutation process or fitness landscapes. As examples, we consider several illustrative cases, including (1) a case where mutations can only be acquired sequentially (Figure 6, Figure 6 figure supplement 2) and (2) a case where mutations occur in a distance-dependent manner in the space of rescaling parameters, in essence making mutations between strains with similar scaling parameters more likely than those with vastly different scaling parameters (Figure 6 figure supplements 3-4).

Similar to the case with selection only, it is difficult to draw general conclusions (e.g. “synergistic combinations *always* accelerate resistance”) that apply across all classes of drug combinations, collateral effects, and mutational structures. The relationship between these features and the evolutionary dynamics is complex, but as these examples illustrate, the framework can be extended to incorporate these features and produce testable predictions. In the discussion, we discuss this point as follows:

“Our goal was to strike a balance between analytical tractability and generality vs biological realism and system specificity. We stress that the predictions of this model do not come for free; they depend, for example, on properties of the dose response surfaces, the collection of scaling parameters, and the specific mutational structure. In many cases, these features can be determined empirically; in other cases, some or all of these properties may be unknown. The evolutionary predictions of the model will ultimately depend on these inputs, and it is difficult to draw general conclusions (e.g. “synergistic combinations always accelerate resistance”) that apply across all classes of drug combinations, collateral effects, and mutational structures. But we believe the framework is valuable precisely because it generates testable predictions in situations that might otherwise seem intractably complex.”

2) Somewhat relatedly, the reviewers had some issues with use of the term "model" to describe the Price equation. While the mathematics and framing themselves are quite responsible, there were some particular issues raised about the framing. I want to especially point you towards the comments from reviewer #2, who explained issues with usage of the term "model" and "prediction" throughout.

We thank the reviewers for their comments, and we have taken additional care to discuss these important, but subtle, distinctions in the revised manuscript. We agree that the manuscript is not merely about the Price Equation, which (as the reviewers point out) is really just an algebraic restatement of generic statistical features of this (and similar) models. Our underlying model formalizes the concept of concentration rescaling as a general description of resistance, a continuous trait that evolves in a manner that depends on the multi-component environment, and that model could indeed be analyzed in the absence of the Price Equation connection. At the same time, we believe the connection to the Price Equation has conceptual value, not only because it links our work with a classical result in evolutionary biology, but also because it provides a convenient form for disentangling the effects of collateral resistance and drug interactions (at least in the stated limits). We now include the following statement in the Results, just after the brief derivation of the Price Equation for our model:

“In what follows, we will sometimes casually refer to Equation 5 as a "model", but it is important to note that the Price Equation is not, in and of itself, a mathematical model in the traditional sense. Instead, it is a simple mathematical statement describing the statistical relationship between variables, which are themselves defined in some underlying model. In this case, the mathematical model consists of a collection of exponentially growing populations whose per capita growth rates are linked by scaling relationships. Equation 5 – the Price Equation – does not include additional assumptions, mechanistic or otherwise, but merely captures statistical relationships between those model variables. We will see, however, that these relationships provide conceptual insight into the interplay between collateral effects and drug interactions.”

In addition, we now consistently refer to the “model” rather than the “Price Equation” when describing our simulations and results (e.g. in the figure captions).

3) There are several issues with figures, figure labels, captions and descriptions that require the author's attention. As this paper does hinge on an understanding of the particulars, it is especially important that figures clearly communicate the results. Please see the individual reviewer's comments for specifics.

Thank you for these suggestions. The particulars indeed make this a challenging story to tell, and we agree that some parts of the initial presentation added unnecessary confusion. We have revised all the figures (and captions) in question in an effort to clarify, and in some cases simplify, the presentation. Most notably, we have completely remade Figure 7 (previously Figure 6) while correcting minor errors and moving additional details to the SI.

Reviewer #1:This study utilizes the price equation in order to describe evolution of drug resistance, recognizing the cutting-edge problem of predicting the evolution of resistance in light of collateral effects and drug-drug interactions.The manuscript offers a formalism that cleverly integrates theory across paradigms, from quantitative genetics to the evolution of antibiotic resistance. As an intellectual exercise, this is courageous, the type of integrative thinking necessary in the modern iteration of evolutionary medicine.

We thank the reviewer for the positive evaluation.

I am not, however, of the mindset that the Price equation is so important a framing that its invocation/application to a different paradigm (antibiotic resistance) is especially interesting by itself. Said differently, I am not personally intrigued by the a study that is successful at connecting antibiotic resistance to the Price equation. Alternatively, I am much much more intrigued by the possibility that the Price equation might add a richer and more nuanced overall perspective to cutting-edge problems in the evolution of antibiotic resistance.

The reviewer makes a good point. We agree that the manuscript is not merely about the Price Equation, but instead about formalizing the concept of concentration rescaling as a general description of resistance as a continuous trait that manifests itself in environmentally-dependent ways. Note that this is quite distinct from previous applications of the Price Equation, which typically cast resistance as a binary trait (0 or 1) and where the primary goal is predicting mean trajectories for the frequency of resistance in a population. In addition, we believe the connection to the Price Equation within our rescaling model has conceptual value, not only because it links our work with a classical result in evolutionary biology, but also because it provides a convenient form for disentangling the effects of collateral resistance and drug interactions (at least in the stated limits).

And I can say that, on the balance, the manuscript was successful in offering a new take on relevant problem.Also, this study carries out its methods carefully, and its formalisms are well-written. The essential idea is that the vagaries of drug interactions and collateral effects can be collapsed into a formalism that considers both when predicting the trajectory of drug resistance, is a fascinating and important one.The study is especially transparent (refreshingly so) about the limitations of its scope. And while this transparency is a legitimate positive of this story, I did find myself concerned about the small number of cases that theory proposed in this study applies to. For example, the Price equation as applied in the study applies to selection on standing variation, and not for the step-wise acquisition of resistance that fitness landscapes are often constructed to study. The authors argue that this type of nuance can be engineered into the formalism, which is not obvious to me (I'm not suggesting that it isn't true, only that it isn't obvious).But that this study has a relatively thin application space does not make the study irrelevant. For the outlined problem space, the arguments appear sound.I must say that given the current state of the problem of antimicrobial resistance in 2021, sometimes featuring complex, highly dynamic, epistastic fitness seascapes, for example, I was hoping for framework that could be used for fitness landscapes as they are currently constructed and studied.On the balance, however, I found the study to be well-constructed, written, and the formalisms to be correct. The main concerns regarding the scope might be a matter of preference. And even I must recognize that responsible theoretical biology often requires a narrow scope in order to generate formalisms. The idea here is that from this contribution, the community can then extend the study to other such systems and data sets.

We thank the reviewer for recognizing our efforts to be transparent, for highlighting the value of the formalism – and also recognizing the difficulty of developing theoretical biology with broad scope – and for the generally positive appraisal. The revised version extends the formalism to include mutations, which allows us to incorporate empirical constraints on the mutational pathways linking different phenotypes (similar, though not identical, to the information contained in fitness landscapes). The trade-off with this added complexity is, of course, the need for additional assumptions, and while we have focused on simple illustrative examples (e.g. sequential mutations, Figure 6 and Figure 6 figure supplement 2, and distance-dependent mutations in parameter space, Figure 6 figure supplements 3-4) we believe they illustrate that the approach is flexible and amenable to new empirical constraints as they become available. We discuss these points in the revised manuscript, and (most notably) we also added the following to the discussion:

“Our goal was to strike a balance between analytical tractability and generality vs biological realism and system specificity. We stress that the predictions of this model do not come for free; they depend, for example, on properties of the dose response surfaces, the collection of scaling parameters, and the specific mutational structure. In many cases, these features can be determined empirically; in other cases, some or all of these properties may be unknown. The evolutionary predictions of the model will ultimately depend on these inputs, and it is difficult to draw general conclusions (e.g. “synergistic combinations always accelerate resistance”) that apply across all classes of drug combinations, collateral effects, and mutational structures. But we believe the framework is valuable precisely because it generates testable predictions in situations that might otherwise seem intractably complex.”

I would like the authors to consider, and possibly cite, if they are relevant, more recent papers by Pamela Yeh on this topic:Tekin, Elif, Pamela J. Yeh, and Van M. Savage. "General form for interaction measures and framework for deriving higher-order emergent effects." Frontiers in Ecology and Evolution 6 (2018): 166.Tekin, Elif, Van M. Savage, and Pamela J. Yeh. "Measuring higher-order drug interactions: A review of recent approaches." Current Opinion in Systems Biology 4 (2017): 16-23.Tekin, Elif, et al., "Enhanced identification of synergistic and antagonistic emergent interactions among three or more drugs." Journal of The Royal Society Interface 13.119 (2016): 20160332.

Thank you for these suggestions. We have cited all 3 papers, which focus on higher-order (more than 2) drug interactions and therefore may be useful for extending our framework to higher dimensions.

There are some small issues with the figures that need to be resolved:I found Figure 6 to be complicated. I would suggest the authors re-generate it with the goal of explaining it to someone not well-versed in the system.

We have simplified and remade Figure 6 (now Figure 7), which we agree was overly complicated in the initial draft.

I had some issues with the legends in Figures 4 and 5. Ensure that they describe the information in the figures properly.

We have clarified the legends of Figures 4 and 5, which were indeed confusing in the initial draft.

Reviewer #2:This paper examines the interaction between pharmacological and evolutionary factors that mediate the emergence of resistance to combination antimicrobial therapy. The authors introduce and demonstrate the utility of a mathematical equation that describes how both (a) the pharmacological interaction between two drugs (e.g. synergy vs antagonism) and (b) the interaction between the fitness effects of mutations that confer resistance to one drug or the other (or both), jointly determine the direction of selection in the presence of two-drug therapy. They show that the equation they describe follows the same form as a two-trait version of the classical "Price equation" in population genetics. They also examine a "weak selection" limit (when evolution proceeds by selection of a sequence of strains each with only a slight change in resistance level from the previous). In this limit their equation has an even more intuitive physical meaning as movement along an weighted version of the gradient of the drug's dose-response curve, where the weighting reflects the nature of the cross-resistance profile. Many examples of the implications of this equation for different drug interactions and different fitness landscapes are presented. The results of this paper unite findings from quite a few previous studies in a common framework. For example, the authors show how to understand (i) when resistance to one drug in a combination will be more strongly selected than resistance to the other and how this direction of selection can change based on the relative levels of two drugs in a combination – even if total inhibition is the same, (ii) when drug antagonism (vs synergy) is morel likely to slow down the rate of adaptation, and (iii) how the relative ordering and frequency of cycling between two drugs in a combination impacts the rate of resistance emergence.

We thank the reviewer for the concise summary and for recognizing the intuitive value of the framework and its potential to unify findings from several previous studies.

It is important for readers to understand that the Price equation – and thus by extension the result in this paper – is not really a mathematical "model" that includes assumptions about the underlying biological processes and makes predictions about potential experimental results. Instead, the Price equation is a mathematical truth that simply involves decomposing certain statistical relationships between variables and expressing them in an intuitive way. Roughly, in this case the Price equation describes how the average level of resistance in a population of microbes will change over time depending on the frequencies of different strains in the population and their resistance levels, including the correlation between resistance to each drug. Thus, this paper is primarily purely theoretical/mathematical work.

We agree with this point, and we have now been careful to clarify this distinction. Most notably, we added the following to the Results section, just after the introduction of the Price equation (copied from response to Evaluation Summary, above):

“In what follows, we will sometimes casually refer to Equation 5 as a "model", but it is important to note that the Price Equation is not, in and of itself, a mathematical model in the traditional sense. Instead, it is a simple mathematical statement describing the statistical relationship between variables, which are themselves defined in some underlying model. In this case, the mathematical model consists of a collection of exponentially growing populations whose per capita growth rates are linked by scaling relationships. Equation 5 – the Price Equation – does not include additional assumptions, mechanistic or otherwise, but merely captures statistical relationships between those model variables. We will see, however, that these relationships provide conceptual insight into the interplay between collateral effects and drug interactions.”

In addition, we now consistently refer to the “model” rather than the “Price Equation” when describing our simulations and results (e.g. in the figure captions).

The authors do apply their equation to some real-life drug profiles and fitness landscapes using dose-response curves for two-drug antibiotic combinations to calculate the direction of selection for different dose combinations. Later in the paper, they do also apply their equation as a "model" to predict entire evolutionary trajectories, by additionally assuming that they can define a priori all the available resistance mutations and ignore any additionally generated mutations. Then, they show that the level of resistance selected after a specific time as predicted by their equations agrees relatively well with the results of in vitro experiments containing these same mutations. This is actually quite a surprising and nice result.Overall this paper provides a useful new framework for understanding drug resistance evolution and provides multiple examples of insights offered by this framework. They provide a nice connection between the mainly mathematical nature of their results and scenarios involving antibiotic treatment and resistance evolution in the lab. The introduction of this paper is very comprehensive and summarizes a huge body of work on the evolution of drug resistance. The discussion is equally thorough and mentions most of the caveats I describe here.Although the authors of the paper focus on bacterial infections and antibiotic resistance, I think the results are much more general, and likely also apply to multi-drug therapy to viral infections (e.g. HIV, HCV), parasitic infections (e.g. malaria), and even anti-cancer therapy.While not a weakness that is specific to this paper, it is important for readers to understand that a Price equation only describes the act of selection acting on existing genetic variation in a population, and does not describe the act of generating that variation (i.e. mutations). Thus, while this approach can be helpful for understanding the direction and speed of selection in one vs two-drug combinations of different types, it cannot be used to understand the overall risk of resistance emergence, since the rate at which mutations are generated may also differ. For example, it often takes two separate mutations to generate resistance to two drugs with different mechanisms of action, so resistance to two-drug therapy is often slowed by the rate of generating resistant mutations. In addition, the rate at which mutations are generated during therapy depends on the population size over time, which might differ along different selective paths described by the Price equation for different drug combinations. Therefore, there are some important results from previous work related to factors that accelerate or inhibit multi-drug resistance which cannot be understood in the framework presented in this paper.

Thank you for these points. We have now included additional discussion of the limitations of the model, potential applications and, more importantly, we now illustrate several examples where mutational dynamics are included. We copy below our response to Point 1 in the Evaluation Summary (above):

In the revised manuscript, we include a new section (including 1 additional figure, Figure 6, and several SI figures, S8-S10) that incorporates mutation. Under these conditions, the Price Equation includes an additional term that depends on a mutation matrix that describes the probability of mutation between any pairs of strains (Equation 13). Under some conditions (e.g. when mutations occur with uniform probability between the ancestor cells and each mutant), the effects of mutation are qualitatively similar to those of selection (Figure 4 and Figure 6 figure supplement 1). On the other hand, mutational matrices can be chosen to reflect known features of the mutation process or fitness landscapes. As examples, we consider several illustrative cases, including (1) a case where mutations can only be acquired sequentially (Figure 6, Figure 6 figure supplement 2) and (2) a case where mutations occur in a distance-dependent manner in the space of rescaling parameters, in essence making mutations between strains with similar scaling parameters more likely than those with vastly different scaling parameters (Figure 6 figure supplements 3-4).

Similar to the case with selection only, it is difficult to draw general conclusions (e.g. “synergistic combinations *always* accelerate resistance”) that apply across all classes of drug combinations, collateral effects, and mutational structures. The relationship between these features and the evolutionary dynamics is complex, but as these examples illustrate, the framework can be extended to incorporate these features and produce testable predictions. In the discussion, we discuss this point as follows:

“Our goal was to strike a balance between analytical tractability and generality vs biological realism and system specificity. We stress that the predictions of this model do not come for free; they depend, for example, on properties of the dose response surfaces, the collection of scaling parameters, and the specific mutational structure. In many cases, these features can be determined empirically; in other cases, some or all of these properties may be unknown. The evolutionary predictions of the model will ultimately depend on these inputs, and it is difficult to draw general conclusions (e.g. “synergistic combinations always accelerate resistance”) that apply across all classes of drug combinations, collateral effects, and mutational structures. But we believe the framework is valuable precisely because it generates testable predictions in situations that might otherwise seem intractably complex.”

One assumption that the authors use in order to get a simplified Price-equation-like result is that the effect of a drug resistance mutation on the dose-dependent effect of a drug on microbial growth can be described by a single parameter, which just shifts the drug concentration to a lower value. This is equivalent to saying that a mutation simply alters the IC50 of the drug. However, many studies have shown that in general mutations have at least one and often two other effects on dose-response curves: firstly, they have a fitness cost, meaning that the maximum fitness in the absence of drug is lower, and secondly, they can also alter the slope of the dose response curve. It is unclear how these effects change the equation the authors have derived.

Thank you for this suggestion. We now include additional discussion of this point in the discussion.

The impact of the authors approach on studying resistance evolution in any particular system depends on the regime that system is in. If resistance tends to evolve by selection slowly acting on an extremely diverse population containing strains with many different resistance levels, then this approach will have the most utility. Alternatively, if resistance evolution tends to be limited by rare mutational events that produce strains with large differences in resistance levels from the parental strain, then this approach will have less use.To summarize, the most important caveat for readers to understand is that the mathematical expressions that come out of this paper should not be said to be able to "predict" evolution, since they say nothing about the availability of possible resistance mutations or the rate at which they could be generated.

We agree with these points. We have added brief discussions of these points in the new section on mutations as well as in the discussion (as we have noted in the previous responses).

Concerns and suggested changes/clarifications:– It is a bit misleading to call the Price Equation a mathematical model. It's not a model in the way that word is generally used in biology – it doesn't descibe a mechanism or make any assumptions, and it's not subject to experimental verification. It's just a mathematical statement about a statistical relationship among variables, based on the definition of those variables and basic laws of calculus. The authors should be very clear about this throughout the paper.– The parts of the paper that are more like a "model" are the parts where you use the Price Equation to describe an evolutionary trajectory, assuming a particular distribution of mutants from standing genetic variation and ignoring additional mutations, or, when you assume a weak selection regime– Overall the authors should be more careful about the use of the word "model" and claims about "predictions" throughout

Thank you for these important suggestions. In the revision, we have been careful to distinguish between the assumptions of our model (e.g. concentration rescaling, statistical features of collateral effects, or particular mutational structures) and the Price Equation, which the reviewer rightly points out is simply a restatement of statistical relationships emerging from the underlying model. We have also been more careful in our use of “prediction”, which can indeed be misleading in this context. Please see also our responses to point 2 in the Evaluation Summary above.

– I'm not sure about Equation 2 – 5 requires the assumption that the effect of a mutant can be described by a rescaling of drug concentrations. Instead, I think you could replace the α parameters with another parameter r for example that simply means "resistance level" but does not specify how the "resistance level" changes the shape of the dose-response curve.

The reviewer is correct that selection need not occur only on the drug concentration, but can instead alter any feature of the dose response curve, and we now discuss this briefly in the discussion.

– Equation 5 actually is only half of the regular Price equation. It is the "selection" component but is missing the "transmission" component. The authors should be clearer about what processes relevant to the evolution of drug resistance they are ignoring that allow them to set this second term to zero.

As described above, the revised manuscript now includes a new analysis including the “transmission” (in this case, mutation) terms. Please see our response to point 1 in the Evaluation Summary above.

Comments on Figures and results:Figure 2 – What are the units of the axes on all these graphs?a) I don't understand this statement in the caption: "We then simulated an external condition where the combination drug concentration is held constant at 0.6 of the maximal concentration of drug 1 and 0.1 of the maximal concentration of drug 2.". There is no such thing as a "maximal" drug concentration … you can always keep adding more and more drug. It would be clearer just to state the drug level used in absolute terms. Or maybe you mean the "highest tested drug level"?

Thank you for the suggestion. We have now rewritten the caption for clarity.

Figure 3 – Some of the results shown here are a little confusing and hard to understand. In the black scenario, there are few available resistant strains with lower α, so we expect selection to proceed in the direction of Drug 2/β.. that agrees with the results in B. But for the blue case where, despite lots of available α but little available β, the trajectory in B seems relatively symmetrical in both drugs … why? Also I was surprised that the case of collateral sensitivity (green) wasn't the worst case … but I realized it's a bit misleading because the four scenarios don't only differ in the nature of the correlations in resistance levels, but also in the magnitude of the variance in α or β and hence the best possible available mutation. I think that it would be a fairer comparison if for the green and magenta scenarios the variance was narrower around the lines of perfect correlation/anticorrelation.

Indeed these results can be counterintuitive. In short, they arise from a combination of the collateral effects and the drug interaction. While the reviewer is correct that the four different cases are not a precise “apples to apples” comparison, the seemingly strange results do make sense in light of the rescaling hypothesis (and are actually not driven by the somewhat smaller differences in correlation parameters generating the collateral profiles). For example, the black trajectory in B is not dominated by resistance to drug 2, which is what one would expect from the collateral effects alone. The reason is that additional resistance to drug 2 does not offer significant growth advantages at this dosage combination due to the drug interactions. Intuitively, one can see this by noting that a rescaling of drug 2 concentration would correspond to moving downward on the contour plot. But moving downward at this location would not change growth dramatically because the rescaled concentration would fall near the same contour as the original drug concentration (i.e. the growth contour itself is nearly vertical at this location). We have added a brief discussion of this point in the main text and also included the specific covariance parameters in the caption.

Figure 4 – This is nice figure that really summarizes the combined effects of drug interactions and cross-resistance profile. However this figure also summarizes the major limitations of this approach. The early slopes of each growth rate vs time curve do show how these factors interact to determine selection strength. However, the fact that the maximum growth rate varies by curve is really just an artifact of the arbitrary and limited available mutants the authors have created for selection to act on. In reality, mutations are continually and sequentially generated over time, and this rate of generation of mutants also impacts the increase in growth rate over time and might remove any "maximum" resistance level that is achievable.

The reviewer makes a good point: the steady state behavior in the no-mutation limit is likely to be determined by growth of a single mutant (or strictly, a series of mutants that fall along a single growth contour), and it is therefore highly dependent on the extremes in the set of scaling parameters. In practice, new dynamics are likely to arise on these long time scales, and the model is expected to fail as mutational processes (and potentially a breakdown of the scaling assumption, in general) become increasingly relevant. The upshot is that the no-mutation model is useful over a limited time horizon, though fortunately, that timescale is similar to that observed in many lab evolution experiments aimed at understanding drug interactions and collateral effects. Perhaps more importantly, we have also added a mutational process to the model in the revised manuscript. While the set of available mutants must still be assigned up front, different mutational structures can be chosen to mimic longer-time processes (e.g. by making mutation rate dependent on the distance between mutants in scaling parameter space, the evolutionary trajectory may proceed through increasingly fit phenotypes). We have also included new supplemental figures that illustrate the connection between the time of observation and potential saturation of the dynamics, though in a slightly different context (Figure 5 figure supplement 4; Figure 7 figure supplements 1-2).

Figure 5 – Are these predicted trajectories using the weak selection limit or the full equations without this additional assumption?– I'm assuming there was some fixed time limit that selection was allowed to proceed for in these experiments and that it was the same in all cases?– B: is the y axis absolute IC50 or change in IC50? The axes titles just say IC50 but the figure caption says "relative change in IC50". Also it would be helpful if the baseline IC50 (ie for the wild type strain) where shown on these graph as well.

We have clarified the caption to note that (1) these simulations are from the full equations and (2) these experiments were performed for a fixed time period of 72 hours, and (3) IC50 for each drug is measured in units of the ancestral strain IC50 (so they are indeed normalized IC50s). The wild type IC50 is therefore 1 for each drug (this was not clear in the original version because we didn’t properly indicate that the IC50 is normalized).

Finally, I don't understand how the results for the TGC IC50 agree with the results in A. From B, the IC50 looks totally flat with TGC concentration. But from A, it looks like there is a large increase in TGC resistance (increase in TGC IC50) for high TGC concentrations, but little TGC resistance for low TGC concentrations.

We have now revised the caption to hopefully clarify these points. In panel A, we show a collection of 100 trajectories at each condition, with each trajectory based on a uniformly sampled collection of approximately 10 parameter pairs. The black x marks indicate the mean value (across all trajectories for a given dosage combination) at the end of the simulation. In B, we plot these final-time average values (normalized IC50 corresponds to the reciprocal of the corresponding scaling constant) as a function of TGC concentration in the selecting environment (which here is just a proxy for the different external dosage combinations, the different conditions, as you move left to right on the contour). The discrepancy you note at high TGC concentrations arises from the fact that the trajectories with large increases in TGC are relatively rare and therefore contribute little to the trajectory average.

– C: I don't understand what the y axis is here. If this is the increase in growth rate, then why is it highest for low TGC concentrations, yet both panels in B show no change in IC50 to either drug at low TGC concentrations?

The axis shows increase in growth rate, which we have clarified in the caption. At low TGC concentrations, panel B shows that the change in TGC IC50 is minimal (TCG scaling parameter near 1) while the change in CIP IC50 is larger (it increases by a factor of roughly 3-4). In terms of rescaling, this trend makes intuitive sense: a vertical rescaling (increased resistance to CIP) leads to an increase in growth for conditions at low TGC, where the rescaling moves the effective concentration into a region of higher growth. But for higher TGC concentrations, such a vertical concentration rescaling would not be expected to have much effect on growth because a vertical shift in effective drug concentration does not lead to large deviations from the original growth contour (and in fact experiments in this regime did not select for large increases in CIP resistance). In words: TGC has a strongly suppressive effect on CIP (strongly antagonistic drug interaction). As a result, low TGC concentrations select for CIP resistance while higher TGC concentrations do not, even when the CIP concentrations are similar.

And any idea what could be causing the systematic difference between the predicted and observed changes in growth rate?

This is an interesting question, and we don’t know the answer. One potential explanation is that the time chosen for the simulation (72 hours) does not match the true time over which selection was taking place in the experiment. The experiment was performed for 72 hours, but various environmental variables (fluctuations in temperature, density-dependent changes in growth rate, etc) could mean that the evolution occurred, effectively, for more or less than 72 hours. The upshot is that the total number of generations may not match perfectly between model and experiment. We did not attempt to tune this parameter to fit the data, but indeed we can decrease the systematic error by choosing a different value of T (but we prefer to show the model without fitted parameters). It is also possible that the systematic error arises from other features, such as uncertainty in the estimates of the wild-type growth surface. We have now explicitly noted and briefly discussed the systematic discrepancy in the main text.

Figure 6 – This figure was quite confusing, it needs more annotation on the figure itself (not just written in the caption), and some panels are unnecessarily small (with unnecessary white space between them). For panels H and K, are there better axis limits that could be used so that 95% of the figure is not all one color? Also I found it is hard to see in this figure any of the differences between the outcomes of the two different orderings that are described in the text.

We have replaced this figure (now Figure 7) with a simplified version. Our hope is to highlight one way that the framework can be extended to make predictions in an otherwise intractably complex scenario. However, the previous figure was unnecessarily confusing. In the new figure, we choose a simpler example than in the previous version, and we focus on a smaller set of the dynamics.

Reviewer #3:In this work, Gjini and Wood create a model to simulate evolutionary dynamics of antibiotic resistance arising as a response to drug combinations. They demonstrate that evolution to these drug combinations is a variation of the Price equation and show that some evolutionary paths experimentally correspond to a weighted gradient descent on the growth landscape.The work appears mathematically sound, though we have concerns about the scope and generality of the results. In particular this work appears to rely fundamentally on the assumption that drug resistance acts as a linear rescaling of the two-drug dose response curve. We believe this work would be strengthened by making it more clear what, beyond the mathematical description of the rescaling hypothesis they cite extensively, this work adds to the literature. Since the model appears to also require an a priori description of the resistance characteristics of every possible accessible mutant, which typically is not known, it isn't clear the utility of the model.

Thank you for these suggestions. In our view, there are two main advances. First, we incorporate multiple features of the evolutionary process (collateral effects, drug interaction, and, in the new version, mutational structure) into a model that makes testable predictions. As with all models, the outputs are only as good as the inputs, and we can not use the model to predict resistance evolution without those inputs (e.g. the drug interaction and statistical properties of the collateral effects). However, when those properties are known, or when we can make reasonable assumptions about them, the model allows us to make predictions that would not otherwise be possible and to systematically investigate the impacts of different phenomena (e.g. collateral effects vs drug interactions). Second, we believe the approach has considerable conceptual value. It provides a direct link to the Price Equation, which connects the work with classical approaches in population genetics. Perhaps more importantly, the Price Equation formalism allows us to mathematically disentangle effects of collateral resistance and drug interactions, at least in the stated limits, in a way that highlights how the geometry of the response surface interacts with statistical properties of the available mutants.

We also note that in the new version, we now incorporate mutation between different phenotypes, which significantly extends the scope of the model (see Figure 6 and Figure 6 figure supplements).

Please also see our responses to points 1 and 2 in the Evaluation Summary (above).

We also copy below a (somewhat related) response to a suggestion from Reviewer 1:

We thank the reviewer for recognizing our efforts to be transparent, for highlighting the value of the formalism, and also recognizing the difficulty of developing theoretical biology with broad scope, and for the generally positive appraisal. The revised version extends the formalism to include mutations, which allows us to incorporate empirical constraints on the mutational pathways linking different phenotypes (similar, though not identical, to the information contained in fitness landscapes). The trade-off with this added complexity is, of course, the need for additional assumptions, and while we have focused on simple illustrative examples (e.g. sequential mutations, Figure 6, and distance-dependent mutations in parameter space, Figure 6 figure supplements 3-4) we believe they illustrate that the approach is flexible and amenable to new empirical constraints as they become available. We discuss these points in the revised manuscript, and (most notably) we also added the following to the discussion:

“Our goal was to strike a balance between analytical tractability and generality vs biological realism and system specificity. We stress that the predictions of this model do not come for free; they depend, for example, on properties of the dose response surfaces, the collection of scaling parameters, and the specific mutational structure. In many cases, these features can be determined empirically; in other cases, some or all of these properties may be unknown. The evolutionary predictions of the model will ultimately depend on these inputs, and it is difficult to draw general conclusions (e.g. “synergistic combinations always accelerate resistance”) that apply across all classes of drug combinations, collateral effects, and mutational structures. But we believe the framework is valuable precisely because it generates testable predictions in situations that might otherwise seem intractably complex.“

– This paper models evolution in the presence of physiological interactions between antibiotics but not evolutionary interactions. The fundamental assumption that evolution of resistance scales the surface but does not change its shape, which is central to the findings, is true in some cases, though violated for many mutations (the authors cite numerous papers on cross-resistance and collateral sensitivity that are examples of this).

We agree that the rescaling assumption will not always hold, and we discuss these limitations extensively in the discussion. We also discuss the fact that the model does not incorporate interactions between cells (e.g. cooperation and competition), it is an avenue for future work, we hope! We would like to point out, however, that cross-resistance and collateral sensitivity in and of themselves are not indications that the scaling assumption is violated (testing this would require measuring the full dose response surface in the mutants and determining whether a simple rescaling of the two axes is sufficient to link the two surfaces, that is, whether the shape of the surface changes, not merely the axes scales). There is considerable experimental support suggesting the scaling assumption provides a reasonable approximation, at least for conditions of a typical lab evolution experiment, but indeed the assumption will not always hold, and we do note that important limitation.

– While the authors describe it as "straightforward" to include discrete mutational events instead of a small population of equal numbers of every potential mutant in the model, it is not clear that this is the case. For example, a population exposed to an above-MIC concentration of a bacteriostatic drug will no longer generate additional mutants, and so the standing diversity before drug application matters critically. As it stands the model is really about differential growth and not evolution per se, since neither mutation nor complete extinction can occur.

We have incorporated mutation in the revised manuscript (see response to point 1 in the Evaluation Summary, above). But the reviewer makes an important point: the model accounts only for differences in growth rates between different strains. It does not account for changes in total population size, though these effects would be particularly important as the population nears extinction. The model also neglects demographic stochasticity, which could play a significant role in some regimes. We’ve added a short discussion of these points in the discussion.

– It isn't clear the experiments are representative of nontrivial behavior predicted by the model. For example in figure 5 no adaptation to TGC was seen, so it seems the only real evolution was rescaling CIP, which wouldn't yield the suppression-induced selection against resistance they then discuss. These results could be entire an artifact of the selection being entirely on CIP and not at all on TGC. The evolution experiments in Figures S3 and S4 seem much more compelling, though.

We found this example quite compelling, and at first we were puzzled by your comment. But then we realized that there was a bug in the code that plotted the experimental results, which obscured the most interesting features of the experimental data (they were still present in the model). Correcting this plotting error also led to an improved agreement between experiment and model. Thank you for your comment, which helped us avoid a silly mistake (the bug was apparently introduced during an internal revision of the figure prior to submission; it wasn’t there in our original internal drafts but showed up in the version we submitted, and we simply missed it).

With that bug now fixed, hopefully it is clear why we find the example compelling. Selection for CIP resistance occurs only for sufficiently low concentrations of TGC, even in cases where inhibitory concentrations of CIP are present. For example, evolution in the far left condition (TGC~0, CIP~0.2) leads to a 2-3 fold increase in CIP IC50 (panel B) and significant growth adaptation (panel C). However, evolution at the 3rd condition from the right (TGC~0.04, CIP~0.2) shows very little resistance to CIP, despite the fact that a similar concentration of CIP was used. The fourth condition from the right is even more striking, as there the concentration of CIP is even higher (about 0.35) yet little CIP resistance is observed.

– If it is indeed straightforward to add mutational events, this would make the model appreciably more realistic.

We have now incorporated mutation in a new section of the manuscript. Please see response to point 1 in the Evaluation Summary (above).